# A Practical and Analytical Comparative Study of Gel-Based Top-Down and Gel-Free Bottom-Up Proteomics Including Unbiased Proteoform Detection

**DOI:** 10.3390/cells12050747

**Published:** 2023-02-26

**Authors:** Huriye Ercan, Ulrike Resch, Felicia Hsu, Goran Mitulovic, Andrea Bileck, Christopher Gerner, Jae-Won Yang, Margarethe Geiger, Ingrid Miller, Maria Zellner

**Affiliations:** 1Centre for Physiology and Pharmacology, Medical University of Vienna, 1090 Vienna, Austria; 2Immunology Outpatient Clinic, 1090 Vienna, Austria; 3Proteomics Core Facility, Clinical Department of Laboratory Medicine, Medical University of Vienna, 1090 Vienna, Austria; 4Department of Analytical Chemistry, Faculty of Chemistry, University of Vienna, 1090 Vienna, Austria; 5Joint Metabolome Facility, University of Vienna and Medical University of Vienna, 1090 Vienna, Austria; 6Institute of Medical Biochemistry, University of Veterinary Medicine Vienna, 1210 Vienna, Austria

**Keywords:** top-down proteomics, 2D-DIGE, bottom-up proteomics, shotgun proteomics, proteoforms, post-translational modification (PTM)

## Abstract

Proteomics is an indispensable analytical technique to study the dynamic functioning of biological systems via different proteins and their proteoforms. In recent years, bottom-up shotgun has become more popular than gel-based top-down proteomics. The current study examined the qualitative and quantitative performance of these two fundamentally different methodologies by the parallel measurement of six technical and three biological replicates of the human prostate carcinoma cell line DU145 using its two most common standard techniques, label-free shotgun and two-dimensional differential gel electrophoresis (2D-DIGE). The analytical strengths and limitations were explored, finally focusing on the unbiased detection of proteoforms, exemplified by discovering a prostate cancer-related cleavage product of pyruvate kinase M2. Label-free shotgun proteomics quickly yields an annotated proteome but with reduced robustness, as determined by three times higher technical variation compared to 2D-DIGE. At a glance, only 2D-DIGE top-down analysis provided valuable, direct stoichiometric qualitative and quantitative information from proteins to their proteoforms, even with unexpected post-translational modifications, such as proteolytic cleavage and phosphorylation. However, the 2D-DIGE technology required almost 20 times as much time per protein/proteoform characterization with more manual work. Ultimately, this work should expose both techniques’ orthogonality with their different contents of data output to elucidate biological questions.

## 1. Introduction

A central goal of proteome research is to understand the composition and function of the proteins in a biological sample. The completion of the human genome sequencing project in 2003 and the surprising identification of only about 20,300 distinct genes [1] made the one-gene-one-protein dogma [2] even more unlikely. Thus, the size of the human proteome is still debatable, with estimates ranging from 20,000 to several million different proteins and their proteoforms in the literature [3]. These facts indicate that much of the complexity created by biological machinery is at the level of different variants of the respective proteins and is not based on gene diversity [4]. Variations in a protein can occur as a result of different concentrations, genetic mutations, and alternative splicing of DNA-RNA transcripts. Further subsequent changes can result from proteolytic cleavage and numerous covalently linked chemical functional groups on dedicated “vulnerable” amino acids, which are then termed post-translational modifications (PTMs). To date, approximately 400 different PTMs are known in biology (http://www.unimod.org, accessed on 3 September 2022), the most common of which are lysine acetylation, C- and N-terminal cleavage [5], phosphorylation, methylation, glycation, lipidation, and ubiquitination, which dynamically modify proteins throughout their lifespan. PTMs have essential regulatory properties, such as switching a protein from its inactive to its active or thereafter inactivated state, or regulating a protein’s half-life following ubiquitination or acetylation, thus defining its functional property in a cell and tissue-specific context that ultimately determines the resulting cellular phenotype and its biological significance [6].

The term “proteoform” was defined in 2013 to provide a uniform definition for all these different possible protein variations [4]. A specific designation for proteins with PTMs has previously been defined with the term “protein species” [7]. To record all these regulatory processes at the protein level, an exact quantification of the proteins together with all their proteoforms is necessary. The number of proteoforms that a protein can have is theoretically impossible to predict. Based on the two-dimensional gel electrophoresis (2D-GE) data, it was assumed that each protein has, on average, three different proteoforms in eukaryotes [8]. A more recent work, the Blood Proteoform Atlas [9], found about 17.5 proteoforms per human gene using highly complex technical MS-based top-down proteomics. However, it is noteworthy that this MS-based top-down proteomics analysis mainly recognises proteoforms with a molecular weight of less than 20 kDa, of which lysine acetylation (32.9%) and the C- and N- terminal cleavage (30.6%) are the two most common [5]. Thus, an immense variety of proteoforms is currently not sufficiently considered in analytics, and no analytical method can fully decode the entire proteomic diversity of a complex biological sample.

The general strategy pursued in proteomics is to compare related samples from different states (e.g., healthy vs. diseased/exposed/treated) since differences in their proteome should reflect the particular state of a biological sample. In the two main different proteomics approaches, gel-based and gel-free, the quantification of biological differences is done at different steps in the workflow: in gels immediately after separation and at the protein spot level. Identification of the respective proteins is not yet required for this step. In the gel-free LC-MS approach, also called shotgun proteomics, it is essential to know protein identity before quantification, as peptides need to be related to each other and to their parent proteins; only then is protein quantification possible. Hence, gel-based proteomics usually only identifies proteins with different abundance, while LC-MS has to identify all detected proteins.

Today, most of the proteome analysis is performed with label and label-free shotgun proteomics. Label-free proteomics is more commonly used in large-scale biological studies because it requires less manual work, can be automated to some extent, and requires only minute amounts of the sample [10]. Overall, it is, therefore, faster and cheaper and enables quantitative high-throughput sample analysis [11]. However, the limited stability of the instrument components, liquid chromatography (LC), and mass spectrometry (MS) aggravate reproducibility. For shotgun analysis, intact proteins are enzymatically disassembled into peptides (e.g., primarily by trypsin) to facilitate separation by reversed-phase liquid chromatography, followed by directly coupled analysis in mass spectrometers. Intact proteins can no longer be examined, hence the term bottom-up proteomics. All measured peptides are reassembled *in silico* into the putative proteins or protein groups for quantitative profiling of the respective proteomic sample [10,11]. This evaluation method must therefore go back to the outdated “one gene, one protein dogma,” and the results describe only the qualitative and quantitative composition of so-called “theoretical” or “canonical proteins” in the samples [11]. Thus, this peptide-centric approach has lost all essential qualitative and quantitative information from the corresponding proteoforms. For shotgun analysis experts, this protein inference is a significant and well-known problem [12,13,14]. Interestingly, this significant disadvantage of shotgun proteomics is still almost wholly ignored in numerous analytical applications. [15].

The dynamic complexity of a proteome is currently best demonstrated by the top-down method 2D-GE. [11,16,17,18,19,20,21]. The term top-down proteomics in 2D-GE refers to intact proteins and their intact proteoforms being detected with this method [12,13,19,20,21]. This methodology allows the qualitative separation of intact proteins and their proteoforms based on their physico-chemical properties, which are determined by their respective isoelectric point (pI) and molecular weight (MW) [22,23,24,25]. Since each proteoform has a specific pI and MW, they can be readily separated and detected using 2D-GE. A protein’s mobility in the pI and MW dimensions can be altered, for example, by proteolytic cleavage, phosphorylation and the substitution of an amino acid due to an SNP, etc. However, in order to decode a proteoform´s identity, the protein has to be excised from the gel, digested and finally analysed by MS. Although this procedure can be automated to a certain extent, it is still very time-consuming and thus a limiting factor in the application of this technology.

Two-dimensional (2D) fluorescence difference gel electrophoresis (2D-DIGE) is currently a widely used variant of quantitative 2D-GE electrophoresis analysis with the best quantitative precision. Direct labelling of a protein’s lysine with differentially spectrally resolvable cyanine fluorescent dyes (e.g., 488 nm/520 nm, 532 nm/580 nm, 633 nm/670 nm, 736 nm/760 nm) prior to 2D fractionation of proteoforms into 2D spots enables the simultaneous analysis of two to four different proteomic samples in one analytical 2D run. In this 2D analysis, one specific dye (e.g., Cy2) is often reserved for an internal standard (IS). This IS is usually a pool of all samples measured in all gels of the respective analytical study, allowing for perfect qualitative and quantitative comparability between the different 2D-GE runs and all the separated proteome samples [26].

With all these technical advantages and disadvantages, a systematic comparison of both techniques is required for their synergetic use to deepen knowledge in biological investigations. So far, however, these two proteomic methods have rarely been used in combination to investigate the proteomic composition of biological samples as comprehensively and deeply as possible. In these few available studies, both methods were used to increase the probability of finding as many proteins [27] or condition-dependent protein abundance changes as possible in the respective biological samples but without taking care of proteoforms [28,29,30,31,32,33]. Two other studies combined these proteomics technologies to characterize potentially robust method-independent biomarkers, such as in liver tumour samples [34] or frozen-thawed curled octopus [35]. Another study used both methods to determine which proteins co-occur in different cell types and can be detected using various proteomics technologies. Thus, this protein repertoire should serve as quality control for the sensitivity of the respective proteomics experiment [36]. However, none of these studies attempted to improve the profiling of proteins and their proteoforms by combining these two proteomics technologies. We could only find two publications in which the presence of different proteoforms of the respective proteins was deliberately and application-relatedly included in the parallel analysis by evaluation of the biological sample using 2D-DIGE and LC-MS/MS shotgun [37,38].

In addition, despite the utmost care in the analysis, the measured qualitative presence and quantitative amounts of proteins and their proteoforms from the same sample can also differ from gel to gel and from MS to MS run. A crucial parameter for the reliable detection of qualitative and quantitative changes in the abundance of proteins and their proteoforms of a biological sample is the evaluation of the system-specific technical and biological variation, which thus represents the total variation. Surprisingly, no study has yet directly compared these qualitative and quantitative properties, as well as the possible synergistic properties of both proteomics methods in a practical experiment.

Therefore, this study focuses on the technical variability and orthogonality in the respective technical and biological data outputs of top-down or bottom-up proteomics analysis. For this purpose, identical technical and biological replicates were analysed using the 2D-DIGE and label-free shotgun technologies. Subsequently, coefficients of variation (CV) were determined from the respective quantitative data and comparatively evaluated, particularly considering the aspects of proteoforms and phosphorylation in specific biological examples. In parallel, all these strengths and limitations of the two proteomics techniques were considered and discussed together with the aspect of workload and time.

## 2. Materials and Methods

### 2.1. Cell Culture

DU145 human prostate carcinoma cells (ACC 261) were purchased from Leibniz Institute DSMZ (Braunschweig, Germany) and cultivated at 37 °C in RPMI 1640 (Gibco, Carlsbad, CA, USA), supplemented with 10% heat-inactivated foetal bovine serum (Sigma, St. Louis, MO, USA), 100 U/mL penicillin and 0.1 mg/mL streptomycin (Fisher Scientific, Waltham, MA, USA) at 37 °C in a humidified incubator in an atmosphere of 5% CO_2_, 20% O_2_ and 75% N_2_. DU145 cells were subsequently propagated in T75 vessels and seeded in 6-well plates, grown to 90% confluency, washed two times with PBS and dry plates were stored at −80 °C until cell lysis for proteomic analysis.

### 2.2. Protein Isolation from DU145-Prostate Cells for Top-Down and Bottom-Up Proteomics

For the preparation of cell lysates, 6-well microplates were allowed to reach ambient temperature (15–20 °C) to prevent precipitation of 2D-DIGE buffer (7 M urea, 2 M thiourea, 4% CHAPS, 20 mM Tris, pH 8.7), aliquots of 400 µL 2D-DIGE buffer were added and cells were scraped and collected in 1.5 mL tubes, solubilised for 2 h at 4 °C at 850 rpm to facilitate complete protein solubilization and centrifuged at 15,000× *g* for 10 min to remove insoluble material. The total protein concentration of the lysates was quantified by using a Bradford Coomassie Plus kit (Pierce Thermo Scientific, Rockford, IL, USA).

Three biological replicates were generated from three different cell passages for parallel top-down and bottom-up proteomics to evaluate the total (biological + technical) variation of the particular method. A pool of these biological replicates was made for an IS sample commonly used in 2D-DIGE analysis to standardize the 2D spot signals across different gel runs. The six technical replicates of the 2D-DIGE analysis were all performed with this IS sample, labelled with Cy2, Cy3 and Cy5, in three paired gel runs (Figure 1A) of the pH 4–7 and pH 6–9 range, respectively. This IS sample was also used for label-free shotgun analysis to evaluate the technical and total variation as well as the performance in detecting phosphopeptides. Both gel-based and gel-free proteomic methods were used to analyse these samples’ technical and biological replicates in parallel.

### 2.3. Gel-Based Top-Down Proteomics by 2D-DIGE

The gel-based method was performed by 2D-DIGE, followed by an MS protein identification. Twelve µg of technical or biological replicate was minimally labelled with fluorescent cyanine dyes (5 pmol of CyDyes per µg of protein; GE Healthcare, Uppsala, Sweden), as already described in [39]. Three technical replicates were labelled with either Cy3, Cy5 or Cy2. For the acidic protein range, immobilised pH gradient (IPG) strips (24 cm, pH 4–7, GE Healthcare, Uppsala, Sweden) were passively rehydrated for 11 h with rehydration buffer (7 M urea, 2 M thiourea, 4% CHAPS, 70 mM dithiothreitol (DTT), 0.5% pH 4–7 ampholyte (Serva, Heidelberg, Germany)) mixed with a total of 36 µg of alternatively Cy-labelled sample. Isoelectric focusing (IEF) was performed on a Protean I12 IEF unit (Bio-Rad) with gel side down until 30 kVh were reached. For the alkaline protein range, IPG-strips (24 cm, pH 6–9, GE Healthcare, Uppsala, Sweden) were soaked in rehydration solution (7 M urea, 2 M thiourea, 4% CHAPS, 150 mM DTT, 2% ampholyte pH 6–10 (GE Healthcare, Uppsala, Sweden)) prior to isoelectric focusing (IEF). Samples were applied by cup loading on the acidic side and DTT (325 mM) loading on the cathode of the IPG strip. IEF was performed on an IPGphor unit (GE Healthcare, Uppsala, Sweden) with gel side up until 30 kVh were reached. After IEF, IPG-strips were then each equilibrated with equilibration buffer (buffer 1: 1% DTT, 50 mM Tris-HCl pH 8.68, 6 M urea, 30% glycerol and 2% sodium dodecyl sulfate (SDS), for 20 min; buffer 2: 2.5% iodoacetamide (IAA), 50 mM Tris-HCl pH 8.68, 6 M urea, 30% glycerol and 2% SDS, for 15 min) under slow shaking. Each of the IPG-strips was transferred on 11.5% polyacrylamide gel (26 × 20 cm, 1 mm gel thickness) and sealed with low melting agarose sealing solution (375 mM Tris-HCl pH 8.68, 1% SDS, 0.5% agarose). The SDS-PAGE was performed using an Ettan DALTsix electrophoresis chamber (GE Healthcare, Uppsala, Sweden) under the following conditions: 35 V for 1 h, 50 V for 1.5 h and finally 110 V for 16.5 h at 10 °C. The gels were scanned with a resolution of 100 µm using a Typhoon 9410 laser scanner (GE Healthcare, Uppsala, Sweden) at excitation/emission wavelengths of 532/670 nm (Cy3), 633/670 nm (Cy5) and 488/520 nm (Cy2). It was a time interval of two months between the analysis of the technical and biological replicates.

### 2.4. 2D-DIGE-Based Characterisation of Phosphorylated Proteins by λ-Phosphatase Treatment

The IS sample was taken for the characterisation of the phosphorylated protein spots in the 2D-DIGE map of the DU145 cell line. Two aliquots of 90 µg of each sample (IS) were mixed with 5 µL of 10% SDS and vortexed for 10 s. Then, samples were filled up to 500 µL with a reaction mix containing 2 mM MnCl_2_, 5 mM DTT, 1× lambda-phosphatase (λ-PPase) buffer, and dH_2_O. One sample was incubated overnight (14 h) with 100 units of λ-PPase (30 °C, under gentle agitation). Subsequently, all samples (±λ-PPase) were precipitated with TCA containing 80 mM DTT for 1 h at 4 °C, pelleted at 20,000× *g* for 20 min at 4 °C, and washed four times (20,000× *g* for 10 min, 4 °C) with acetone containing 20 mM DTT. The protein concentration was again determined to calculate the 2D-DIGE buffer volume to solubilise the samples with a concentration of 2.5 µg/µL after TCA-precipitation.

### 2.5. Protein Identification of 2D Spots via LC-MS/MS

For MS-based identifications, 250 µg unlabelled proteins were separated by the same 2D-DIGE equipment that was used for the fluorescently labelled samples described above. Proteins were visualised by MS-compatible silver staining [40]. Protein spots of interest were excised manually from the gels, de-stained, disulphide was reduced, and afterwards, derivatised with iodoacetamide, and the proteins were digested with a concentration of 12.5 ng/µL sequencing grade modified trypsin (Promega, Madison, WI, USA). Electrospray ionization (ESI) quadrupole time-of-flight (QTOF; Compact, Bruker, Billerica, MA, USA) coupled to an Ultimate 3000 Nano HPLC system (Dionex, Sunnyvale, CA, USA) was used for LC-MS/MS-based identification of spot digests. In this system, a PepMap100 C18 trap column (300 μm × 5 mm) and PepMap100 C18 analytic column (75 μm × 250 mm) were used for reverse phase (RP) chromatographic separation with a flow rate of 500 nL/min. The two buffers used for the RP chromatography were 0.1% formic acid/water and 0.08% formic acid/80% acetonitrile (ACN)/water with a linear gradient for 90 min. Eluted peptides were then directly sprayed into the MS, and the MS/MS spectra were interpreted with the Mascot search engine (version 2.7.0, Matrix Science, London, UK) against the Swissprot database (564,277 sequences, released in January 2021) and the taxonomy was restricted to homo sapiens (human; 20,397 sequences). The search parameters were used with a mass tolerance of 10 ppm and an MS/MS tolerance of 0.1 Da. Carbamidomethylation (Cys), oxidation (Met), phosphorylation (Ser, Thr and Tyr), acetylation (Lys and N-term) and deamidation (Asn and Gln) were allowed with 2 missing cleavage sites. The Mascot cut-off score was set to 15, and proteins identified with two or more peptides were considered [41]. Furthermore, a protein was considered as reliably identified only when its associated peptide counts were at least five times higher than those from other protein identifications of this 2D spot.

### 2.6. Gel-Free Bottom-Up Proteomics by Label-Free LC-MS/MS Shotgun

For shotgun proteomic analysis of DU145 cells, 50 µg protein lysates of the IS and the three biological replicate samples in 2D-DIGE buffer were subjected to methanol-chloroform-water (MCW) precipitation to remove detergents and salts. In brief, protein samples were diluted to 100 µL with dH_2_O, 400 µL methanol was added, vortexed for 1 min, 100 µL of chloroform was added and vortexed and finally, 300 µL of dH_2_O was added and samples were vortexed. Samples were centrifuged at 14,000× *g* at 4 °C for 15 min. The upper phase was removed, and the protein-interface was precipitated by the addition of 300 µL methanol. Samples were vortexed and left for 15 min at −20 °C, followed by centrifugation as before. The protein pellet was washed with methanol, air-dried and dissolved in 0.1% RapiGest SF (Waters, Milford, MA, USA) in 50 mM triethylammonium bicarbonate (TEAB), reduced by DTT (5 mM, 30 min at 60 °C) and alkylated in the dark by IAA (15 mM, 30 min, room temperature). Samples were digested using mass-spec grade Trypsin/Lys-C mix as suggested by the manufacturer (Promega, Madison, WI, USA) overnight; digests (16 h) were stopped by the addition of trifluoroacetic acid (TFA) (1% final concentration). Peptides were desalted and concentrated following the stage-tip protocol by Rappsilber et al. [42] using 3 layers of reversed-phase Empore Octadecyl C18 solid phase extraction disk stacked in a 200 µL pipet tip and stored at −20 °C until MS analysis. Peptides were eluted twice with 10 µL acetonitrile (ACN) and 10 µL 0.1% TFA, dried in a SpeedVac and solubilised in 12 µL peptide resuspension buffer (2% ACN and 0.1% FA). The technical and biological replicates of the tryptic peptide DU145 samples were separated by a 70 min gradient on a C18 µPAC (µ-Pillar-Arrayed-Column, PharmaFluidics, Ghent, Belgium) mounted on a nano RSLC UltiMate3000 (Thermo Fisher Scientific, Waltham, Massachusetts, USA) separation system. Peptides were detected as described earlier [43,44]. In brief, peptides (2 µL) were introduced into the nano electrospray source (ESI) after the UV cell, and the ionization was performed using a steel needle with a 20 µm inner diameter and 10 µm tip. Needle voltage was set to 2 kV in positive mode, and the top 10 ions were selected for MS/MS analysis (fragmentation). The resolution was set to 70,000 for full MS scans, a mass range of 350–1700 *m*/*z*, ions with single charge were excluded from MS/MS analysis and fragmented ions were excluded for 60 s from further fragmentation. During each run, the lock mass ion 445.12002 from ambient air (polysiloxane) was used for real-time mass calibration. Raw MS/MS files were analysed with MaxQuant version 1.6.0.1 with default settings for “label-free quantification” (LFQ), and match between runs was enabled, variable modifications were set to oxidation (M), acetyl (protein N-term) and phospho STY [44] against the human proteome (http://www.uniprot.org/proteomes/UP000005640_9606, (accessed on 30 September 2018), version from September 2018). LFQ and match between runs were enabled, and variable modifications were set to oxidation (M), acetyl (protein N-term) and phospho STY.

The label-free quantification approach is based on the computational methodology described by Jürgen Cox et al. 2014 [45], where the intensities of the precursors (MS1) are used to quantify across the technical and biological replicate samples. This data output of the label-free LC-MS/MS shotgun analysis method was used, as shown in Figure 2, to document the workload and technical and total variation of this bottom-up method compared to the 2D-DIGE analysis. It is also important to mention that there was a time interval of several months between the analyses of the technical and biological replicates.

### 2.7. Direct Label-Free-Shotgun Proteomics-Based Characterisation of Phosphorylated Proteins by λ-Phosphatase

To evaluate the performance for detecting phosphopeptides and their corresponding technical variation (CV_tech_) in label-free shotgun measurement, 10 µg aliquots of the IS standard sample were subjected to λ-PPase treatment as described above or not and subjected to MCW-precipitation. Proteins were solubilised in 8 M urea and 2 M thiourea, reduced and alkylated. Afterwards, sequential digestion was made by LysC (2 h), followed by trypsin (overnight, 16 h). Desalted peptides were diluted in 25 µL loading buffer (2% ACN, 0.05% TFA) and subjected to LC-MS/MS analysis as follows. To this end, 5 µL of peptide sample were injected into the Dionex Ultimate3000 nanoLC system (Thermo Fisher Scientific, Waltham, MA, USA). For sample pre-concentration, a pre-column (2 cm × 75 µm C18 Pepmap100; Thermo Fisher Scientific) run at a flow rate of 10 µL/min using mobile phase A (99.9% H_2_O, 0.1% FA) was used. Chromatographic separation was performed on a 25 cm × 75 µm Aurora Series emitter column (IonOpticks, Fitzroy, Australia) by applying a flow rate of 300 nL/min and using a gradient of 8% to 40% mobile phase B (79.9% ACN, 20% H_2_O, 0.1% FA) over 95 min, resulting in a total LC run time of 135 min per sample. For mass spectrometric analyses, the timsTOF Pro MS (Bruker) equipped with a captive spray ion source was used. The capillary voltage was set to 1700 V, and the MS/MS spectra were generated in the Parallel Accumulation-Serial Fragmentation (PASEF) mode with a moderate MS data reduction applied. The scan range (*m*/*z*) from 100–1700 for recording the MS and MS/MS spectra was applied. The mobility range was set to 1/k0 from 0.60–1.60 V.s/cm^2^ and the ramp time and accumulation time were set to 100 ms. All experiments were performed with 10 PASEF MS/MS scans per cycle, leading to a total cycle time of 1.16 s. Furthermore, the collision energy was ramped as a function of increasing ion mobility from 20 to 59 eV, and the quadrupole isolation width was set to 2 Th for *m*/*z* < 700 and 3 Th for *m*/*z* > 700. All samples were analysed as technical replicates. MaxQuant version 2.0.3.0 (Computational Systems Biochemistry, Max-Planck Institute for Biochemistry, Martinsried, German) was used to analyse Bruker d.folders with default settings and analytical replica set at single fractions, LFQ and match between runs was enabled, variable modifications were set to oxidation (MP), acetyl (protein N-term), deamidation (N) and phosphor (STY) and the *fasta* database was the same as described above. MaxQuant result outputs (proteinGroups.txt, Phospho(STY)Sites.txt, evidence.txt and peptides.txt) were analysed and visualised in Perseus version. 1.6.14.0 (Computational Systems Biochemistry, Max-Planck Institute for Biochemistry, Martinsried, German).

### 2.8. One and Two-Dimensional Western Blot Analysis

For one-dimensional Western Blot (1D-WB), a total of 12 μg of the urea-dissolved DU145 protein extract was mixed with a sample buffer (150 mM Tris-HCl pH 8.68, 7.5% SDS, 37.5% glycerol, bromophenol blue, 125 mM DTT) to obtain a final volume of 20 μL. These samples were boiled for 4 min at 95°C and centrifuged for 3 min at 20,000× *g*. The samples were then run in an 11.5% SDS gel (50 V, 20 min and 100 V, 150 min), separated and blotted (75 V, 120 min) onto a polyvinylidene difluoride membrane (PVDF; FluoroTrans RW, Pall, East Hills, NY, USA). The molecular weight separation and the transfer to the membrane of the DU145 protein samples were monitored with a protein molecular weight marker (PageRuler, Prestained Protein Ladder, Life Technologies Limited Inchinnan, Renfrew PA4, UK). For detection of the blotted proteins, the total protein on the membrane was stained using ruthenium-(II)-tris-(bathophenanthroline disulphonate) (RuBPS; dilution 1:100,000 overnight at 4 C; Sigma-Aldrich St. Louis, MI, USA).

For two-dimensional Western Blot (2D-WB) analysis, 30 µg of the urea-solubilised Cy5-labeled proteins were separated by isoelectric focusing on a 7 cm pH 3–10 IPG-strip (GE Healthcare, Uppsala, Sweden) in the first dimension and according to the MW by 11.5% SDS-PAGE, 10 × 8 cm, (50 V for 20 min and 100 V for 150 min). Then, proteins were semidry-blotted (1.0 A, 25 V, 40 min) onto a polyvinylidene difluoride membrane (PVDF) (FluoroTrans^®^W, Pall, East Hills, NY, USA), followed by scanning with a Typhoon FLA 9500 imager (GE Healthcare, Uppsala, Sweden). Subsequently, membranes were blocked in 5% non-fat dry milk (Bio-Rad, Hercules, CA, USA) in PBS containing 0.3% Tween-20 (PBS-T) overnight at 4 °C.

On the next day, membranes were washed (3× PBS-T for 5 min, each) and incubated for 2 h at room temperature with primary detection antibodies (diluted in PBS-T containing 3% non-fat dry milk) for pyruvate kinase M2 (PKM2; #4053S; Cell Signaling Technology, Boston, MA, USA; 1:1000), 14-3-3 protein γ (YWHAG; #MA1-16587; clone KC21; Pierce, Rockford, IL, USA; 1:10,000), protein disulfide-isomerase A1 (P4HB/PDIA1; #ab2792; clone RL90; Abcam, Cambridge, MA, USA; 1:1000), glyceraldehyde-3-phosphate dehydrogenase (GAPDH; #NBP1-47339; clone 1A10; Novus Biologicals, Littleton, CO, USA; 1:3000), calmodulin (CALM1; #NB110-55649 (EP799Y); Novus Biologicals, Littleton, CO, USA; 1:2000), adenylate cyclase-associated protein 1 (CAP1; #H00010487; clone D01; Abnova, Taipei, Taiwan; 1:1000), eukaryotic initiation factor 4A-I (EIF4A1; # ab31217; Abcam, Cambridge, MA, USA; 1:500), prostaglandin E synthase 3 (PTGES3; #sc-101496; Santa Cruz Biotechnology, Dallas, Texas; 1:1000), transaldolase (H-4) (TALDO1; #sc-166230 Santa Cruz Biotechnology, Dallas, Texas; 1:500), cathepsin B (CTSB; #AF953-SP; R&D Systems, Minneapolis, MN, USA; 1:500), cathepsin D (CTSD; #AF1014-SP; R&D Systems, Minneapolis, MN, USA; 1:500).

After washing (3× PBS-T for 5 min, each), membranes were incubated for 1.5 h at room temperature with the appropriate horseradish peroxidase (HRP)-conjugated secondary detection antibodies diluted 1:20,000 in PBS-T containing 3% non-fat dry milk. After two further washing steps in PBS-T and one in PBS, the immunoreactive bands were developed using SuperSignal Western Femto Maximum Sensitivity Substrate (Thermo Fisher Scientific, Waltham, MA, USA). Chemiluminescence signals were detected on a UVP ChemStudio imager (Analytik Jena, Jena, Germany).

### 2.9. Statistical Analysis

The experimental design and sample sizes are indicated in Figure 1. The images of 2D-DIGE were analysed using DeCyder software (version 7.2, GE Healthcare, Uppsala, Sweden). The standardised abundance (SA) was calculated for protein spot quantifications according to the manual of the DeCyder software [46]. Detailed information about the image analysis was described previously [47]. MaxQuant (version 2.0.3.0, (Computational Systems Biochemistry, Max-Planck Institute for Biochemistry, Martinsried, German)) was used to identify and quantify canonical proteins of label-free shotgun LC-MS/MS runs and phosphorylated peptides and proteoforms.

For the calculation of technical and total variation, the latter consisting of technical and biological variation, the SA values were taken from the 2D-DIGE analysis (Figure 1A), and MaxQuant LFQ protein intensities [45] were taken from label-free shotgun LC-MS/MS runs. The coefficient of variation (CV) was used to calculate the variability of each quantitative analysis system relative to its standard deviation and is presented here as a percentage. Since the SA values of the individual 2D spots are calculated using the normalised volume value from IS of the respective spot, these values no longer contain any information about the volume of the respective proteoform. Therefore, the spot sizes had to be calculated from the normalised spot volume values, which also came from the data output of the DeCyder^TM^ software. Thus, each included spot’s representative spot volume value was calculated from the mean of all technical and biological replicates of the DIGE analysis and the mean of the LFQ values for each included canonical protein range. Mean spot size values from the 2D-DIGE and mean LFQ values from the label-free shotgun analysis are used for Spearman’s rank correlation (r_s_), which was made to determine how well the quantification of these two measurement systems compares to each other. The statistical analyses and graphs were made with GraphPad Prism 7 (GraphPad Software, Inc., San Diego, CA, USA).

## 3. Results and Discussion

This study evaluates the workload, reproducibility, proteomic information output, and synergy of two routine applications of top-down and bottom-up proteome analysis. The prostate cell line DU145 served as the biological sample, although this selection is of secondary importance for the content of this comparative study on basic proteomics technologies. As outlined in the experimental design and workflow in Figure 1, six technical replicates of the 2D-DIGE and six label-free shotgun analysis runs were performed from the same DU145 protein sample to determine the CV_tech_ of the method’s qualitative and quantitative data output. For the evaluation of top-down proteomics, six technical replicates were analysed through three 2D-DIGE gel runs, running one Cy3 and one Cy5 stained replicate on each 2D gel and normalised over the third Cy2 labelled replicate, which represented the IS. Thus, the same sample was separated into nine separate 2D images in these three runs. To cover the entire pH range of gel-based proteome analysis, each replicate was performed in the pH range of 4–7 and 6–9 and assembled into one replicate. To assess the technical reproducibility of a commonly used bottom-up proteome method, a label-free shotgun analysis of six technical replicates from three different digests of the same DU145 samples, each analysed in duplicate by LC-MS/MS, was performed.

Furthermore, cell extracts from three different DU145 cell passages were prepared and analysed in parallel using 2D-DIGE and label-free shotgun runs to measure the total variation (technical + biological) in the proteome of a cell culture system (Figure 1).

**Figure 1 cells-12-00747-f001:**
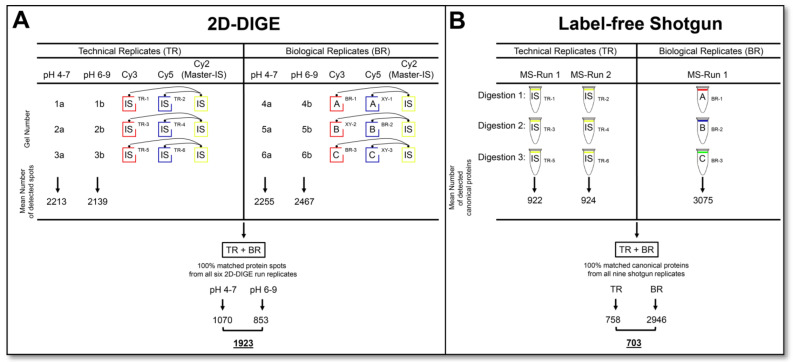
Experimental design and workflow for comparative proteome and proteoform analysis. (**A**) Schematic representation of 2D-DIGE workflow and consistently (100%) matched protein spots (proteoforms) in technical and biological replicates. (**B**) Schematic representation of label-free shotgun workflow and consistently (100%) identified canonical proteins in technical and biological replicates. Detailed statistics on the frequency of the detected protein events are summarised. Abbreviations: TR—technical replicate; BR—biological replicate; IS—internal standard; MS—mass spectrometry.

### 3.1. Time Factor in Proteome Analysis with 2D-DIGE and Label-Free Shotgun

Depending on the proteomics technique, the analysis process requires significantly different amounts of time. Therefore, an important decision criterion for planning a proteomics study is to recognize the time factor in the workflow of the respective proteomics technology compared to an ample yield of well-reproduced and functional, informative data sets. Figure 2 illustrates the time required for a comparative proteomics analysis using 2D-DIGE or label-free shotgun. Good reproducibility of the quantitative data obtained was defined in this comparative study by including only protein events that were found in all analysis runs of the respective proteomics methods.

**Figure 2 cells-12-00747-f002:**
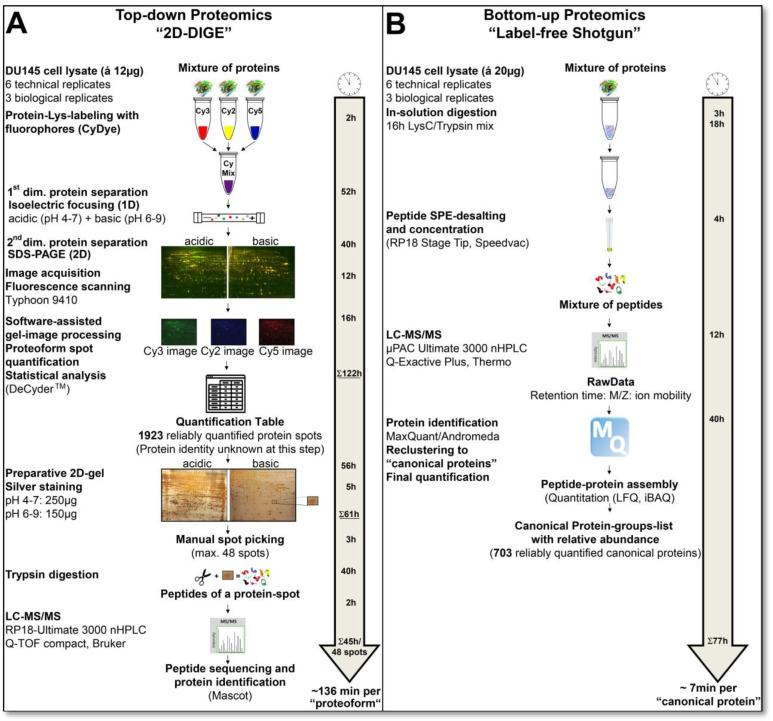
Workflow and time-line of top-down and bottom-up proteome analyses. (**A**) Workflow of top-down gel-based (2D-DIGE) proteome analysis, where proteins/proteoforms are first labelled, separated, computationally detected, quantified and eventually additionally identified by LC-MS/MS. (**B**) Workflow of bottom-up gel-free (label-free shotgun) proteome analysis, where the proteins are first digested, and then the resulting peptides are separated. The detection, identification and quantifications are performed at the peptide level. Abbreviations: 2D-DIGE—two-dimensional difference gel electrophoresis; LC-MS/MS—liquid chromatography mass spectrometry.

In the six technical and three biological replicates of the DU145 cell extracts, a total of 1923 protein spots could be detected over the entire pH range with 100% reproducibility, summarised by six 2D-DIGE runs in the pH range of 4–7 and six associated 2D-DIGE runs in the pH range of 6–9 (Figure 1A). In shotgun-proteomics analysis, 703 canonical proteins with 100% reproducibility in the six technical and three biological replicates of the same set of DU145 samples as used for 2D-DIGE analysis were identified (Figure 1B). Detailed data on the frequency of the protein events detected in each run of the two proteomics technologies are given in Figure 1A,B.

As outlined above and summarised in Figure 2, the preparative and analytical workflows in 2D-DIGE and label-free shotgun proteomics are entirely different. In the top-down 2D-DIGE method, the intact proteome is first separated and then quantitatively analysed. These proteomic working steps of the 12 necessary 2D-DIGE analysis runs took 122 h, from protein labelling of the samples with fluorescent dyes to computer-assisted image analysis. In contrast, bottom-up technologies, such as label-free shotgun analysis, first require tryptic digestion of the samples. In this case, together with the LC-MS/MS runs of these digests, a total analysis time of 77 h in the shotgun approach was necessary for the protein identification and quantitative proteome analysis of 703 canonical proteins. In contrast, after almost twice the working time compared to the shotgun analysis, only the quantitative data of 1923 protein spots were determined by 2D-DIGE without having any protein identifications. In our labour settings, generally preparative 2D silver gels with 250 µg in the pH range 4–7 and 150 µg in the pH range 6–9 of the protein sample are prepared to identify the protein spots in the gel. Optionally higher or medium abundant protein spots may also be directly cut out from the DIGE-gels after staining the gels with an MS-compatible silver stain. The processing time for these preparative gels in these two pH ranges with the DU145 samples was 61 h. Protein spots were manually excised from the gel and tryptically digested for LC-MS/MS analysis in this laboratory setting. With our devices, 48 protein spots for LC-MS/MS can be prepared per run. Thus, the identification of 48 proteins, including manual spot-picking and MS analysis, took about 45 h. Converted per protein spot, representing one proteoform, from cutting to LC-MS/MS analysis with our laboratory equipment, the analysis time is about 1 h to identify one spot in the 2D proteome map of the DU145 cells.

Because of this significant amount of workload and time, all spots of a 2D gel analysis are rarely identified with LC-MS/MS. In order to enable a protein-to-protein comparison with the shotgun data output, 144 different protein spots from the DU145 2D map were first randomly identified in our current study. Therefore, the actual time for this 2D-DIGE proteome analysis, together with the identification of 144 protein spots (=proteoforms), is about 327 h (=13.6 days; 183 h gel work and 144 h spot digestion and MS-based identification) or 136 min per protein quantification and identification for the 2D gel-based method (Figure 2). The label-free shotgun analysis of 703 reliably quantified canonical proteins took 77 h (3–4 days), resulting in a time of 6.6 min (~7 min) per protein quantification and identification. Thus, label-free shotgun analysis was 20 times faster per protein quantification and identification in this specific example. This is one reason why this method has largely replaced 2D technology today.

Apart from this direct comparison of the analysis times of these two proteomics systems, the 2D-DIGE setup requires more manual intervention than the label-free shotgun proteomics. In the 77 h of the label-free shotgun experiment, about 10% is hands-on time for sample preparation, as long as the LC and mass spectrometer run without technical problems. On the other hand, in the 2D-DIGE analysis, 50% of the 327 h of the current study are manual. One highly time-consuming process was the manual picking of the protein spots from the preparative silver 2D gel and their tryptic digestion. Spot-picking and digestion robots developed years ago could process 200–300 protein spots per hour [48]. However, these automated robotic systems have not caught on because they are expensive, and 2D gel-based studies typically do not have enough throughput for efficient use.

### 3.2. Analytical Variations of 2D-DIGE and Label-Free Shotgun Analysis

Essential quality features for a knowledge-generating proteomic analysis of biological samples are sensitivity, specificity, functional insights and the reproducibility of the entire experimental setup. Therefore, it is first and fundamentally important to recognize the total variation of the experimental system, which consists of technical and biological variations. In this chapter, the analytical variability of 2D-DIGE and label-free shotgun analysis is evaluated using parallel measurements of the same DU145 samples.

#### 3.2.1. Qualitative Variations of 2D-DIGE and Label-Free Shotgun Analysis Due to Missing Values

A major challenge in proteomics analysis arises from missing detections of proteins, which reduce the number of comparable proteins in multiple analysis runs. Therefore, these missing values in the respective protein abundance data are one of the main problems in proteomics, as they severely impair the statistical evaluation of 2D gel and shotgun analyses and thus reduce the biological significance [49,50].

For the final comparisons with the shotgun data, the qualitative variability analysis of the current 2D-DIGE runs refers to the six technical and three biological replicates, each proteomic gel data set, composed from the pH range 4–7 and 6–9 (Appendix A). The number of matched protein spots from each 2D-DIGE run to the master gel ranged from 1787–2752 at pH 4–7 and from 1861–3224 at pH 6–9 (data not shown). No significant difference in the number of detected spots was observed between the respective technical and biological replicates. A protein spot was considered reproducible if it was matched in each of the six replicate 2D runs of the respective pH range. Accordingly, 100% reliably matched spots were 1070 for pH 4–7 and 853 for pH 6–9 (Figure 1A). Thus, 1923 protein spots in the 2D-DIGE proteome of the DU145 samples could be quantified with 100% reproducibility in this study (Figure 1A). However, there were more “missing values” at the technically difficult pH 6–9 than at pH 4–7. The reasons for these considerable amounts of missing values in the 2D analysis are that some spots are too weak, are often randomly subdivided by the computer-aided image recognition, some spots are not always equally well separated, or some artefacts in the gels, such as dust, are detected as spots.

The qualitative analysis of variability of the LC-MS/MS runs for the six technical replicates revealed between 823 and 973 different protein identifications per run, and 758 of these proteins were recovered in all these runs. Between 2556 and 2616 proteins were identified in the three biological replicates, of which 2540 were detected in all runs. Across all of these nine replicates, 703 proteins could be found in all of these LC-MS/MS runs (Figure 1B). This significant difference between the detected number of different proteins in the measurement of the technical and biological replicates can be that several months had passed between these two analytical runs, and the LC in the technical replicates did not run in the same quality modus.

The high variability of the detected proteins between the different LC-MS/MS runs shows that “missing values” are also a fundamental problem with the gel-free proteomics technology. Apart from the analytical variability of liquid chromatography in the sufficiently reproducible separation of the peptides, the additional variable of this study setting was the still common “data-dependent-acquisition” (DDA) mode of MS analysis. That is, i.e., the 10 most abundant peptides (top 10 methods) with a certain *m*/*z* within a specific retention time (scan time) are subjected to MS/MS fragmentation within a particular time window (cycle time). They are, therefore, explicitly identified at their molecular amino-acid composition level. Accordingly, this stochastic selection of peptide-precursors in MS1 is intrinsically not 100% reproducible, and neither is the MS/MS fragmentation (MS2). Computational imputation of missing quantitative information at the peptide or protein level by assuming a normality distribution is often used to “compensate” for the missing quantitative information to have sufficient data points for statistical analysis. On the other hand, in MS instrumentation (quadrupole and time-of-flight (ToF) mass analyzers), improved chromatographic peptide-precursor separation technologies such as gas-phase and ion-mobility-based fractionation subsequent to reversed-phase LC realised “inside” of the mass spectrometer, in combination with artificial intelligence-supported computational decoding of detected masses, data-independent-acquisition methods (DIA) are more and more frequently used. In DIA, each precursor-peptide (MS1) is isolated and accumulated (MS2) to yield an amount sufficient to detect peptide fragments (MS3). Thus, the problem of missing value in the shotgun analysis will be sufficiently improved in the future.

A general limiting factor of all proteomics methods is that detecting all proteins in a complex biological sample is impossible. The main reason for this is the very wide concentration range of the various proteins in a complex biological sample. The resolution and staining techniques of the 2D gels are not as sensitive as the well-resolving bottom-up proteomics; therefore, more proteins from the respective proteome will be missing in top-down gel-based proteomics.

#### 3.2.2. Quantitative Variations of 2D-DIGE and Label-Free Shotgun Analysis

Most biological functions and regulations are finally based on quantitative changes in proteins and corresponding proteoforms. However, technical and biological variations in the analysis system can mask these quantitative regulations. Quantitatively accurate proteomics technologies are required to detect as many changes as possible between different biological systems. Therefore, the CV_tech_ of the respective proteomics technique and the biological system’s variability should be known to capture the significant differences of the respective biological question with sufficient sample size and statistical power. To evaluate the respective quantitative variability of the two proteomics analysis systems in the current work, only the protein events detected in all technical and biological replicates of the respective methods were included. Thus, 1923 protein spots were included in evaluating the quantitative variability assessment of the 2D-DIGE analysis and 703 canonical proteins in the label-free shotgun analysis, as shown in Figure 1A,B.

The 2D protein spots have a median quantitative CV_tech_ of 7.6% at pH 4–7 and 8.2% at pH 6–9 (Figure 3). Thus, the median technical, quantitative variation in the current 2D-DIGE analysis of DU145 samples is of the same order of magnitude as we found several years ago with the same method and the same sample size of technical replicates from a human platelet extract [51] and another research group using environmental bacteria [52].

In label-free shotgun LC-MS/MS, the included 703 proteins had a median CV_tech_ of 24% (Figure 3). This higher variability is caused mainly because no internal reference proteins (to control for variations in tryptic digestions) or peptides (to control for variations in retention time or mass deviation) were used. At the same time, in 2D-DIGE, the Cy2-labelled IS sample corrects for technical variations.

Literature for the technical variance of the label-free shotgun analysis generally describes significantly smaller CVs [53,54,55,56]. This discrepancy is mainly due to the log2-transformed quantitative intensity values (i.e., LFQ intensities) commonly used for quantitative differential statistical analysis. However, the incorrect application of this log2 transformation to calculate also the CV leads to significantly lower and wrong CV values [55,57,58,59]. Nonetheless, such log2 data transformations are still taken to calculate the CV from technical or biological replicates [60,61], or occasionally the computational route to CV value calculations remains enigmatic [62].

To examine the biological variation (CV_bio_) of our experimental setup, we analysed three different cell passages of the DU145 cell line with both proteomics technologies, 2D-DIGE, and label-free shotgun. In this case, the CV consisting of CV_tech_ and CV_bio_ is defined as the total coefficient of variation (CV_total_; Figure 3).

The observed CV_total_ between the three DU145 passages was 13% with the 2D-DIGE and 59% with the shotgun-system. Thus, the CV_total_ for both proteomics technologies was higher than the respective CV_tech_ (Figure 3). These results show that three different cell passages of the same cell line exhibit a biological variation in the proteome that contributes to the total variation. However, as with the CV_tech_ (24%), the CV_total_ (59%) was significantly higher than the label-free shotgun analysis of the biological replicates.

With a stable mean quantitative CV_tech_ of the 2D-DIGE system of 7 to 8% in the current as well as in our previous platelet proteomics study, we have a total mean quantitative variation of 13% in different passages of the DU145 samples and a slightly higher total mean variation with 18% was previously observed in the platelet proteome of 20 elderly healthy volunteers [51] as well as in a larger cohort of 238 volunteers [49]. As expected, these observations of the 2D-DIGE system also indicate that the mean quantitative CV_bio_ of the proteome between cell passages of the same cell line is smaller than between platelet samples from different individuals. The higher CV_total_ of the shotgun may also be due to the higher CV_tech_ of these runs and/or a different biological variability of the quantified canonical proteins compared to their different proteoforms.

#### 3.2.3. Comparison of Quantitative Variations in 2D-DIGE and Label-Free Shotgun by Specific Proteins

Therefore, to assess the comparability of quantification as well as the technical and biological quantitative variability of specific proteins from the top-down and bottom-up analysis of the DU145 samples, 144 different protein spots were randomised evenly across pI and MW from preparative silver-stained 2D gels, picked out and analysed by LC-MS/MS. Among these, 138 protein spots were successfully identified. The six other identifications were unassignable and therefore unclear as they identified multiple nearly identical amounts of peptides from different proteins in these “protein spots”. This result also shows one methodological limitation of 2D-GE: Not all proteins/proteoforms can always be sufficiently separated based on their MW and pI. However, since a high-resolution 2D-GE was carried out in this work with two pH gradients (pH 4–7 and pH 6–9) and a broad separation distances of 24 cm in the pI and 20 cm for the MW separation, this problem is reduced, which is shown here with only 4% non-unique assignment of the protein identifications. On average, the unambiguous protein identifications have peptide counts with our 2D-GE separation protocols that are twenty-fold higher than other parallel protein identifications from the respective 2D spot digests.

Significant portions of these clearly identified protein spots were assigned to the same UniProt accession numbers, thus representing the respective proteins’ proteoforms in the examined DU145 proteome. With this random selection, 103 different proteins with a total of 138 different proteoforms were identified in the 2D proteome of the DU145 cells. Eighty-four of the 103 different proteins were also detected with the shotgun analysis and were present with 119 proteoforms (=protein spots) in the top-down 2D-DIGE method (Figure 4, Appendix A). These results show that the 2D method in the DU145 sample alone randomly captures 16% of the proteins with their phenotype-dependent proteoforms qualitatively and quantitatively. So far, however, proteomics studies, primarily using bottom-up technologies, have mainly published statistics on the quantitative changes of canonical proteins of different biological samples. The extent of quantitative variation and condition-dependent biological differences in their corresponding proteoforms has hardly been considered until now.

For the first assessment of how comparable the quantification of bottom-up and top-down methods of the DU145 proteome is, the LFQ values of the canonical proteins were correlated with all corresponding individual protein spot (=proteoform) volumes of the 2D-DIGE analysis and showed only a weak correlation (r_S_ = 0.34; *n* = 119). This feeble quantitative relationship between the two methods is likely due to the shotgun LFQ value of a “canonical” protein being contrasted with several different amounts of its proteoforms. The correlation was improved if the proteoform 2D spot volumes of the respective canonical proteins were also summed (r_S_ = 0.55; *n* = 84). Since it was not validated for this comparison whether all detectable proteoforms were also captured in the 2D-DIGE analysis, a selection of 10 proteins was made, and 2D-WBs in the pH range 3–10 were performed to search for further proteoforms with specific pan antibodies. This proteoform-to-protein comparison of mutually identified entities by 2D-DIGE and shotgun is presented in Table 1.

This selection of specifically detected proteoforms with 2D-WB showed again that a quantitative relationship of LFQ values was only present when the respective 2D spot volumes were grouped into the sum of their canonical proteins (Figure 5B; r_S_ = 0.758; *n* = 10). When this critical factor of several proteoform abundance levels of protein was not taken into account, this correlation substantially decreased (Figure 5A; r_S_ = 0.375; *n* = 34).

### 3.3. An Unbiased Proteoform Exploration Is Only Feasible with 2D-DIGE

Thus, at-a-glance visualisation of individual proteins and proteoforms from a total cell protein lysate, as performed in 2D-GE, is virtually impossible in shotgun analysis, as tryptic digestion reduces the complexity of a sample’s proteoform composition by several orders of magnitude, as shown in Figure 6 and Figure 7. 

In particular, while a protein/proteoform appears as specifiable spot(s) in 2D-GE (Figure 6A), provided that any PTM or alternative (proteolytic) processing event causes a change of the protein´s pI and/or MW, corresponding tryptic peptides of the same protein/proteoform are spread across the entire range of mass-to-charge and retention-time plane (Figure 6B and Figure 7C). Consequently, not only the complexity of a proteome and its quality can be evaluated by a trained eye or image libraries at a glance in 2D-GE, but also a protein´s “flavour” is readily traceable, thereby facilitating the detection of potentially interesting proteoforms characterising environmental, disease and/or drug-treatment response for example. In contrast, such signifying information is virtually lost following the digestion of the biological sample and LC-based peptide separations. After extensive computational analysis, some proteoform information becomes accessible again in bottom-up shotgun proteomics.

Variable modifications on peptides can be partially captured if the correspondingly deduced mass differences on specific amino acids are included in the database-search of the shotgun approach. Similarly, proteoform-specific peptides termed “unique peptides” or “proteotypic peptides”, present the following, i.e., tissue-specific alternative splicing of the corresponding transcript and causing a minute change in pI and/or MW are not readily accessible to quantitation in bottom-up shotgun proteomics. Thus, the relative proportion of such a “non-canonical” modified protein to its “canonical” version in a proteome remains mostly elusive unless proteotypic isotope-labelled peptides are included in the shotgun runs for selective reaction monitoring [63].

### 3.4. The Strength of the 2D-DIGE Methodology Exemplified on PKM2

The following section aims to illustrate the above-described scenario on the basis of a comparative top-down and bottom-up proteome analysis and data output for the glycolytic enzyme pyruvate kinase (PKM). This protein was chosen as an example because it had the most proteoforms from the 2D spots randomly chosen for identification, vividly illustrating the complexity of protein inference. Furthermore, in tumorigenesis, the increasing translational synthesis and level of PKM2 compared to PKM1 is crucial for tumour aggressiveness and has also recently been shown to be diagnostically valuable for prostate cancer progression. This switch to the PKM2 expression is responsible for the Warburg effect of cancer cells [64].

On 2D gels, proteins are often separated in a horizontal chain of their proteoforms, such as spot 15 pI 5.80 and spot 16 pI 6.08 with MW 58 kDa, as shown in Figure 7A. Excision, digestion and MS-analysis confirmed that both of these spots came from the same canonical protein, PKM. In this case, unique proteotypic peptides of the PKM2 protein isoform were detected in both spots along with additional tryptic PKM peptides, clearly showing that PKM2 is the major proteoform in the DU145 cells. In Figure 7B, this sequence coverage map is presented. PKM2 originates from the PKM gene, and PKM1 and PKM2 proteoforms are produced by alternative splicing. The PKM1 and PKM2 proteoforms differ in the canonical sequence only in amino acids 389–433. The MS identification of these two 2D PKM2 spots with MW 58 kDa covers exactly this sequence region to 100%.

In shotgun-proteomics, information about the same protein appears totally different, as illustrated in Figure 7C,D. In the quantitative shotgun data analysis, 29 different PKM peptides, including the PKM2 peptides, were assigned to each other, whereby no intact proteoforms of PKM could be distinguished. From the compilation of these PKM peptides, the quantitative mean (LFQ) of the canonical PKM protein group was calculated (Table 1). The tryptic peptides of PKM2 were distributed throughout the *m*/*z* versus retention time two-dimensional space, and proteoform-specific information for PKM2 (peptide numbers coloured red) was lost like a needle in a haystack (Figure 7C).

Therefore, information on the qualitative and quantitative composition of the PKM2 proteoforms in the DU145 sample cannot be found in the peptide fragments (Appendix A) and is, therefore, also not present in the MaxQuant data output of the shotgun analysis.

To validate the 2D spots’ MS identifications of the PKM2 and to detect possible additional PKM2 proteoforms via a complementary methodology, a 2D-WB was performed in the pH range of 3–10, using a PKM2-specific antibody (Figure 7A). This PKM2 antibody recognised a chain of more than two PKM2 spots at MW 58 kDa between pI 5.80 and 8.20 and a cleavage proteoform of PKM2 at an MW of 42 kDa with a pI of 7.32. Subsequent MS analysis confirmed these other PKM2 proteoforms found immunologically (Figure 7B).

A cleavage product of PKM2 with a similar MW of about 42 kDa was recently found as an enzymatic product from the cysteine proteases cathepsin B and S in pancreatic tumours. This cathepsin-mediated cleavage reduces PKM2 activity and is associated with increased tumour cell proliferation [65]. Therefore, this 42 kDa proteoform of PKM2 can be partially responsible for the Warburg effect and may be a potential biomarker for tumour growth aggressiveness. However, the 42 kDa cleavage product of the current PKM2 proteoform had a protein sequence coverage of 48% with MS analysis, whereby at the N-terminus start, a region of about 32 amino acids is missing and a region of 40 amino acids at the C-terminal end. Two cleavage sites are found for these cathepsins, Q16↓Q17 and Y390↓H391 [62], where only the Q16↓Q17 position can correspond to the amino acid sequence of the current PKM2 cleavage product in the DU145 cells (Figure 7B).

Thus, unbiasedly, only the gel-based top-down proteomics methods could identify the tumour-associated PKM2 as the main PKM proteoforms in DU145 prostate cancer cell lysates. Cleavage of the 42 kDa fragment of PKM2 may be responsible for its reduced enzymatic activity and, thus, in part, for the reduced citric acid cycle-mediated oxidative phosphorylation of the Warburg effect [65].

The presence of this proteolytically processed, relatively unknown proteoform of PKM2 would have been over-looked by a conventional shotgun analysis, as done in the current study, unless specific targeted sample preparation methods, such as the terminal amine isotopic labelling of substrates (TAILS) methodologies, capture novel N-termini following protease-mediated cleavages, would be employed [65,66].

### 3.5. Two-Dimensional Western Blots Are a Useful Tool for the Unbiased Detection of Proteoforms of a Protein

As shown for PKM2 (Figure 7A), screening for additional proteoforms of a given protein by complementary immunological methods, such as antibody-based immunological detection by 1D- or 2D-WB, is valuable as long as specific antibodies are available. In this way, possible proteoforms of a protein can be detected with 1D-and 2D-WBs. For all of these antibodies used for further identification of the respective proteoforms by 2D-WB analysis, their specificity was first validated by 1D-WB with the three biological DU145 replicates of this proteomics study (Appendix A). Further examples are shown in Figure 8. For a conceivable comparison of how these data look in the bottom-up proteome analysis, the shotgun results, which just show protein groups, evidence, peptides and MS/MS data for these selected proteins, are summarised in Appendix A.

In these examples, such as the glycolytic enzyme GAPDH, we found numerous different proteoforms in the alkaline pH region of the DU145 proteome using a 2D-WB (Figure 8A). It is also worth noting that the GAPDH proteoforms show a higher biological quantitative variation than the mean average of the DU145-2D proteome. Because GAPDH is defined as a housekeeping gene, it is believed to have low biological variation. It is therefore used as a normalizing protein for WB analysis to compensate for unevenly applied amounts of protein. Interestingly, we have previously shown in human platelet proteomes that GAPDH exhibits higher biological variation than many other proteins and is, therefore, not a well-suited normalising protein [47].

An example of a protein with only one 2D-detectable proteoform in the DU145 cell lysates was the adapter protein YWHAG. Immunological validation with a specific YWHAG antibody recognised only this spot and no other (Figure 8B). Consistent with this observation, significantly fewer PTMs are reported on the amino acid sequence of YWHAG than for PKM2, with nine in UniProt and three in the Consortium for the Proteoform Atlas (http://repository.topdownproteomics.org/Proteoforms?query=P61981, accessed on 21 October 2022). The electrophoretically clearly separated individual YWHAG spot shows a low quantitative variability with a CV_tech_ of 2.6% and a CV_total_ of 5.5%. A typical loading control, GAPDH, shows a CV_tech_ of 5.6% and a CV_total_ of 20.6%. We have also previously identified this protein with only one proteoform and very low biological variability in the platelet proteome of a large study cohort of 238 subjects [47]. In label-free shotgun analysis, CV_tech_ and CV_total_ of YWHAG were higher at 19% and 44%, respectively, but also here below are the respective mean CVs of all proteins. The typical loading control, GAPDH, shows a CV_tech_ of 14% and a CV_total_ of 54% with the shotgun analysis. Again, the shotgun data could not provide any information about the expected number of proteoforms of GAPDH or YWHAG. 

Further examples of proteins with different numbers of proteoforms in the DU145 proteome are CTSB, EIF4A1, P4HB and CTSD, with their comparative quantitative proteome data output of 2D-DIGE and label-free shotgun analysis (Figure 8 and Table 1). The MS-based LFQ data of these protein samples can show the overall abundance of their canonical proteins at a glance, and pathway analysis with many of their interaction partners can be better done with shotgun data output. However, a protein’s different amounts of potential regulatory proteoforms can currently only be determined with 2D electrophoresis.

Although, in some cases, identifying the PTMs from the respective 2D spots is problematic, if not impossible, since the MS analysis of the respective proteoform rarely achieves 100% coverage. Different concentration of the various proteoforms of a protein also leads to different numbers of MS-identified peptides and, thus, to a differently covered protein sequence.

Despite these analytical challenges in distinguishing the PTM-based differences between the different spot proteoforms of a protein, 2D electrophoresis can be expected to be much more likely to uncover new proteoform-based protein regulations than shotgun analysis. For example, we detected an increased amount of a previously unknown N-terminal cleavage product of the coagulation factor XIII (F13A1) in the platelet proteome from patients with lung cancer [39]. These observations finally indicated that the increased risk of thrombosis in lung cancer could also be related to the altered processing and inactivation of this fibrin-stabilizing coagulation factor, thereby providing a new target for antithrombotic treatment. Moreover, the amount of a proteoform with pI 5.60 of F13A1 correlates positively and another with pI 5.85 negatively with its enzymatic activity. These proteomics results also help elucidate this vital coagulation factor’s previously unknown mechanisms in regulating the enzymatic activity of F13A1.

Another example first discovered using 2D electrophoresis is the major genetic risk factor for Alzheimer’s disease, apolipoprotein E4 [67]. However, it took several years until the single nucleotide polymorphism (rs429358) and thus the exchanged amino acid, cysteine, for arginine at position 112 of this protein could be assigned to this apolipoprotein E4 proteoform [68]. Other proteoform alterations, such as beta-amyloid and hyperphosphorylated tau protein, are also central to Alzheimer’s disease pathology. Thus, we also identified four proteins by a platelet proteomics study using 2D-DIGE as biomarkers for diagnosing Alzheimer’s disease from blood. For three of them, only some of their proteoforms have been modified disease-dependently, such as a splice variant of tropomyosin 1 [69,70].

Therefore, it would be paramount to supplement many shotgun studies and valuable protein information databases, such as the Top-Down Initiative and the Protein Atlas, with 2D gel and 2D-WB proteoform analysis, as presented in the current work. Thus, one would have a quick first unbiased overview of the proteoform profile of the respective proteins. This immunological 2-DE-based fine-tuning of proteoform detection would be an advanced 2-DE database like the USC-OGP 2-DE database introduced and maintained by Angel Garcia [71], which can be found linked in the UniProt database.

### 3.6. Characterization of Phosphorylated Proteoforms with 2D-DIGE and Direct Label-Free-Shotgun Proteomics

Besides information on the workload, reproducibility and quantification of proteoforms, it is also essential to be aware of the different approaches and types of results that can be expected after 2D-DIGE or label-free shotgun analysis when targeting information on PTMs, such as phosphorylation, which are needed to obtain treatment and/or disease-specific biologically relevant information.

According to Uniprot, the protein database which integrates and curates available information on proteins, only 13.0, 31.7 and 36.3% of all proteins which have a serine (Ser), threonine (Thr) and tyrosine (Tyr) are marked as phosphoproteins. Given the highly dynamic nature of these, and most likely all PTMs, the “true” proportion of a phosphorylated protein to its non-phosphorylated one in a biological sample remains largely elusive. Traditional methods in detecting phosphorylated proteins in top-down proteomic approaches are metabolic labelling with the radioactive phosphor (32P and 33P isotopes in tri-, di-, monophospho (A/G/T/C)-nucleosides), phospho-specific fluorescent dyes [72], and the use of phosphor-specific antibodies against the respective phosphosites. Phosphorylation can also be identified at a protein’s exact amino acid position in MS analysis using today’s routine search engine algorithms based on the specific mass difference of the neutral loss of HPO_3_/H_3_PO_4_ (80 and 98 mass units (Da)) detected on amino acids tyrosine (Y), serine (S) and threonine (T), respectively or diagnostic ions (78.959 Da). However, quantitative statistical evaluations of phosphorylated proteoform would require MS analysis to reproducibly detect the particular phosphorylated peptide in a complex tryptic digest of a biological sample or from a much less complex peptide mixture such as a 2D spot digest, but at a DDA setting, this is hardly possible. To increase the sensitivity and reliability for the detection of phosphorylated peptides in bottom-up shotgun analysis, the specific enrichment of phosphopeptides by affinity chromatography, e.g., immobilised metal affinity chromatography or metal oxide affinity chromatography (typically with TiO_2_), is necessary [73,74]. However, the quantitative ratios of phosphorylated to their other non-phosphorylated proteoforms are lost. For a first unbiased look at the sample in question, it is very instructive to investigate what the phosphorylation profiles look like in the original proteome.

### 3.7. Detection of Phosphorylated Proteoforms by the Use of λ-PPase and 2D-DIGE

The use of the λ-PPase, which hydrolyses the phosphate groups of Ser, Thr, Tyr, and His residues [75], is very attractive for the 2D-DIGE system [76]. The loss of a phosphate group increases the pI, resulting in an altered position of phosphoproteins in the 2D map. This effect can be used well with the 2D-DIGE system since the differently fluorescence-labelled original and dephosphorylated samples can be ideally detected in the image analysis. In addition, the information on phosphorylated proteoforms is preserved in the 2D map of the respective biological sample, such as that of DU145 cells, provided that protein preparation and 2D conditions are not changed.

Such an enzymatic cleavage of PTMs from proteins (and peptides) is also occasionally used in shotgun proteomics, i.e., to unmask cysteine reactivity [77], investigate phosphorylation-dependent protein-interactions [78] or to aid detection of glycoprotein-detection [79]. However, PTM-enrichment strategies are much more commonly used in shotgun approaches. As already mentioned, an inherent problem of such enrichment strategies is the loss of stoichiometric information about the different abundance of “native” versus PTM-modified proteoforms.

In this study, we evaluated how λ-PPase treatment of the same DU145 protein lysates assists the detection of phosphorylated proteoforms by 2D-DIGE and phospho-peptide detection in a traditional, “direct” label-free shotgun approach in the original proteome without phosphopeptide enrichment.

Using the 2D-DIGE method, which calculated the ratio of the fluorescent spot signals from the phosphorylated to the dephosphorylated DU145 sample, 81 potentially phosphorylated proteoforms could be detected with a ratio of more than 1.5. This would account for 4% of all protein spots as phosphorylated by this method. The most extensively visible λ-PPase-dephosphorylated protein spots, 13 in number, were selected, excised, in-gel digested, and identified by LC-MS/MS. These proteoforms are indicated in Figure 9A and Appendix A.

Among them is a very well-known phosphorylation substrate, MYL6, the myosin light chain kinase, a critical regulator for tissue contraction. Both a phosphorylated and a non-phosphorylated proteoform of MYL6 could be detected in the 2-DE proteome of the DU145 cell line. In contrast, the phosphorylation profiles of CALM1 and EEF1B2 in this 2D-DIGE analysis show that these proteins could only be detected in the phosphorylated state in the proteome of the DU145 sample. Even with a 2D-WB, only the phosphorylated proteoform of CALM1 could be detected. Both phosphorylated and non-phosphorylated proteoforms could also be detected for the CAP1, PTGES3 and TALDO1. Again, 2D-WB analysis confirmed the presence of phosphorylated and non-phosphorylated spots of these spots (Figure 10A–D).

Furthermore, the 2D profiles of CAP1 and PTGES3 show that accumulating phosphorylation events give rise to these spot chains (reflecting multiple proteoforms). We have previously observed phosphorylation patterns similar to CAP1 for the well-known platelet inactivation marker VASP in the 2D proteome of prostacyclin-treated platelets, using the same method of 2D-DIGE-based analysis of dephosphorylation by λ-PPase. The sequential phosphorylation at different amino acid positions S157 and S256 of the proteoforms causes this phenomenon, visible by their decreasing pI in the 2D-GE. Specific VASP antibodies to detect phosphorylation at S157 and S256 confirmed these observations of λ-PPase treatment [80]. Only recently, it was shown that the amount and the phosphorylation profile of CAP1 are altered in patients with lung cancer and other types of cancer and correlate with the degree of metastasis. Two-dimensional proteomic profiling of CAP1 proteoforms can be helpful in further investigations of the pathological role of CAP1 in cancer [81].

### 3.8. Detection of Phosphorylated Proteoforms by the Use of λ-PPase and Label-Free Shotgun

In contrast to 2D-DIGE analysis, the identification of phosphorylated proteoforms by “direct” shotgun proteomics, that is, without preceding selective affinity enrichment of negatively charged phosphate groups on a proteoform’s peptides to positively charged immobilised metal ions (commonly referred to as immobilised metal affinity chromato-graphy-IMAC), is very limited due to the low abundance of phosphorylated peptides in a complex peptide mixture as described above. To illustrate this, λ-PPase treated protein lysates were digested, and peptides were analysed directly by label-free shotgun on an ion mobility mass spectrometer (timsToF). As expected, while a large number of proteins were consistently identified in DU145 samples (3687 in the absence and 3528 in the presence of phosphatase, respectively, Appendix A), the proportion of phosphorylated proteins was less than 1%, illustrating that detection of phosphorylated peptides assignable to respective proteins by a “direct” shotgun approach is almost circumstantial. 

As summarised in the Venn diagrams (Figure 9B), only 49 versus 40 detected, quantified (numbers of identifications even without intensity are given in parenthesis) and phospho-site localised phosphopeptides on 39 versus 33 proteins were identified and quantitated in samples in the absence and presence of phosphatase, respectively. 

However, the reproducibility of phospho identification was also poor (i.e., only 1 peptide was reproducibly identified (*n* = 4) in non-phosphatase samples, detailed data in Appendix A). Puzzlingly, 28 phosphopeptides belonging to 24 proteins were detected only after phosphatase treatment.

In general, estimating the ratio of the phosphorylated to an unphosphorylated abundance of a protein proves to be complicated in shotgun analysis, regardless of whether direct—as in this study—or phospho enrichment approaches are used. Nevertheless, one of the advantages of the label-free shotgun proteomics over 2D-DIGE is the concomitant identification and quantitation of proteins. To exploit this quantitative information also at the phosphopeptide level, we analysed to which extent the phosphatase treatment is deducible from the reported phosphopeptide intensities of commonly (−/+ λ-PPase) identified phosphopeptides (12 phosphopeptides assigned to nine proteins). We found that phosphopeptide abundance levels identified throughout this direct shotgun approach were, with some exceptions, rather low when not influenced by phosphatase treatment (Appendix A). Only two peptides on two different proteins were found with higher intensity, IFT27 phosphorylated at position 154 and FAM214A phosphorylated at position 879 (Appendix A, Sheet 03). Here, phosphatase treatment impacted the phosphopeptide´s intensity as phosphopeptide counts (*n* = 4) and total peptide counts (7 and 25 for IFT27 and FAM214A in both −/+ λ-PPase, respectively) were identical. As mentioned above, information to which extent (“ratio”) a protein is phosphorylated or not is not directly deducible in this peptide-centric approach, which is in sharp contrast to 2D-DIGE. Using the top-down approach, we could show that the ratio of the phosphorylated to the unphosphorylated MYL6 is 2.7. In contrast, only the phosphorylated proteoform of CALM1 was detectable by 2D-DIGE.

Instead, in shotgun analysis, for modification-specific peptides, such as phosphopeptides, individual peptide intensities are reported, and modified/unmodified peptide ratios are reported only when the corresponding unmodified peptide is recognised in most search algorithms, including MaxQuant. However, this is not always the case, as in this proteomics study. Especially if an enrichment step was included, a proteome and a phosphoproteome measurement are needed for each sample to estimate a ratio of modified/unmodified protein/proteoform. In our direct-shotgun approach, we analysed whether individual phosphopeptide intensities can be normalised to MaxQuant protein-LFQ intensities, obtained by summing up individual protein/proteoform-specific peptide intensities into a protein-abundance value. As depicted in Appendix A, in our direct analysis, the modified peptide intensity contributed to variable degrees, and for some proteins substantially, to the LFQ protein abundance (HMCN1, TRA28, NPM1 and SLIT1, IK in samples without and with λ-PPase-treatment, respectively). 

In contrast, quantification on the basis of iBAQ intensity (the sum of a protein’s measured peptide intensities is divided by the number of theoretically measurable tryptic peptides) reflects the abundance of the phosphorylated versus the unphosphorylated protein much better as iBAQ/p-peptide ratios are mostly lower than 1 in samples without and with phosphatase treatment with one striking exception—nucleophosmin 1/NPM1. This 294AA long protein has 38 trypsin-cleavage sites, theoretically, 12 iBAQ peptides, of which we detected eight, and obviously, the phosphopeptide contributed significantly to the overall iBAQ ratio. These evaluations show that proteoform quantification is more complicated with bottom-up proteomics.

## 4. Conclusions

In summary, this study aims to provide life scientists with valuable, practical examples of the performance of the two established proteome analysis methods, 2D-DIGE and label-free shotgun. This variable quality of data obtained should be considered when planning a comprehensive, unbiased analysis of the functional protein repertoire of a biological sample. 2D-DIGE, a classic top-down proteomics approach, is increasingly considered a “low-throughput” technique compared to seemingly “high-throughput” bottom-up approaches such as shotgun analysis. The latter proteomics technology has obvious advantages, such as immediate protein identification, automation and ongoing advancement of data processing capacity. Despite these breakthrough analytical and technological advances, the underlying biological functions and questions should still guide the analytical approach. As expected, the evaluation of the current work confirmed that the shotgun analysis facilitates a timely and direct proteome profiling over a large abundance range of canonical proteins and enables a high sample throughput. Nevertheless, our data demonstrate that much biological information can be lost due to the higher technical variability and the low probability of reproducible and quantitative stoichiometry detection of proteoforms in a label-free bottom-up approach. Thus, this comparative study shows that the top-down 2D-GE methods remain a robust and highly accurate technique for the unbiased large-scale study of proteoforms and their condition-related qualitative and quantitative changes in biological samples.

Summarising the comparative analytical investigations of our work, we propose that a bottom-up approach is advisable to take a quick, comprehensive look at a biological sample to find changes that are more likely to be transcriptionally or translationally based. However, if modifications at the proteoform level are also expected, such as proteolytic activation, like activation of blood coagulation factors, single nucleotide polymorphisms or short-term modifications in signal transduction pathways, the top-down 2D-DIGE method can be more advantageous. In hypothesis-generating proteomics studies, the alterations in the particular biological samples are often unknown, so it would be ideal to use both methods together in a complementary manner. Using two techniques in coalition can provide interesting new insights into the mutual abundance of proteoforms and their stoichiometric relationships, along with a comprehensive annotated proteome.

## Figures and Tables

**Figure 3 cells-12-00747-f003:**
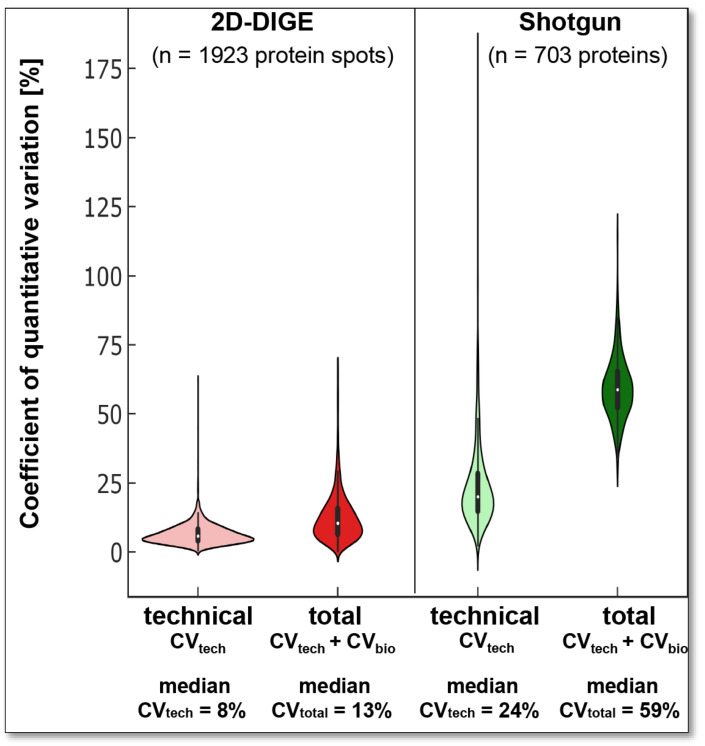
Overall technical and biological quantitative variation in 2D-DIGE and shotgun MS. Violin-blots of technical and total (technical + biological) quantitative variations (CV_total_) in 2D-DIGE and shotgun MS. Abbreviations: 2D-DIGE—two-dimensional difference gel electrophoresis; CV_tech_—technical coefficient of variation, CV_bio—_biological coefficient of variation; CV_total—_total coefficient of variation.

**Figure 4 cells-12-00747-f004:**
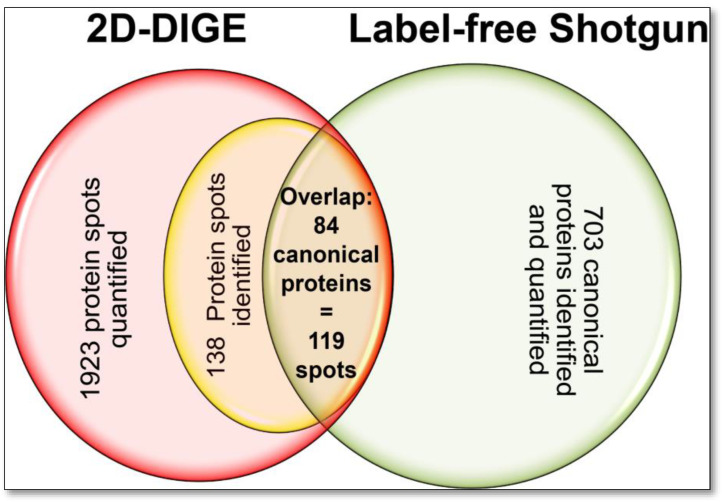
Overview of the number of investigated proteoforms and canonical proteins by 2D- DIGE and label-free shotgun.

**Figure 5 cells-12-00747-f005:**
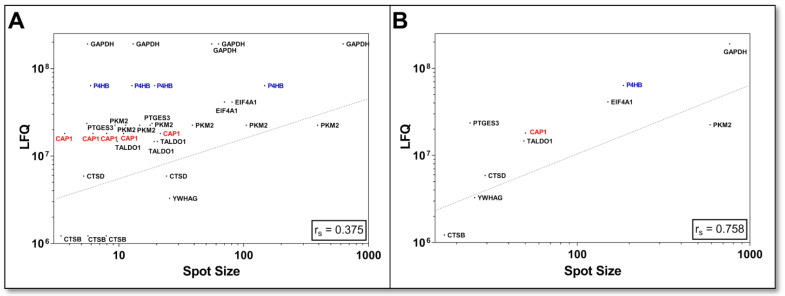
Correlations of quantitative data output from top-down and bottom-up methods. The red and blue proteoforms and canonical proteins are highlighted examples of different quantifications in top-down and bottom-up proteomics. (**A**) The Spearman Correlation factor without summing the abundance of the 2D spot proteoforms (*n* = 34) from the same canonical protein is r_S_ = 0.375, and (**B**) with summing the abundance of 2D spot proteoforms (*n* = 85) from the same canonical protein r_S_ = 0.758.

**Figure 6 cells-12-00747-f006:**
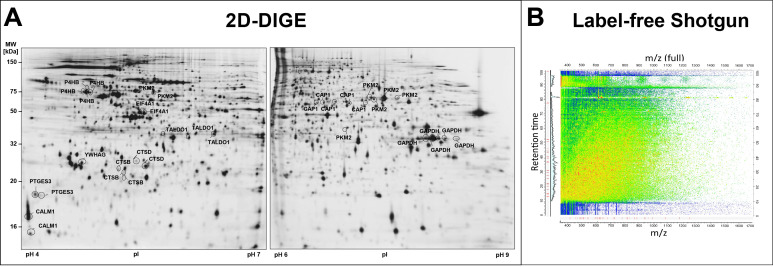
Proteomes at a glance. (**A**) Representative analytical 2D-DIGE gel. Circled protein spots were chosen for proteoform analysis (Table 1). (**B**) Representative heatmap of peptides separated by retention time and mass-to-charge. Centroid-peak view of all identified peptides, red marks at retention time, and *m*/*z* axis indicate peptides assigned to an individual protein (i.e., PKM). Abbreviations: 2D-DIGE—two-dimensional difference gel electrophoresis; MW—molecular weight; pI—isoelectric point.

**Figure 7 cells-12-00747-f007:**
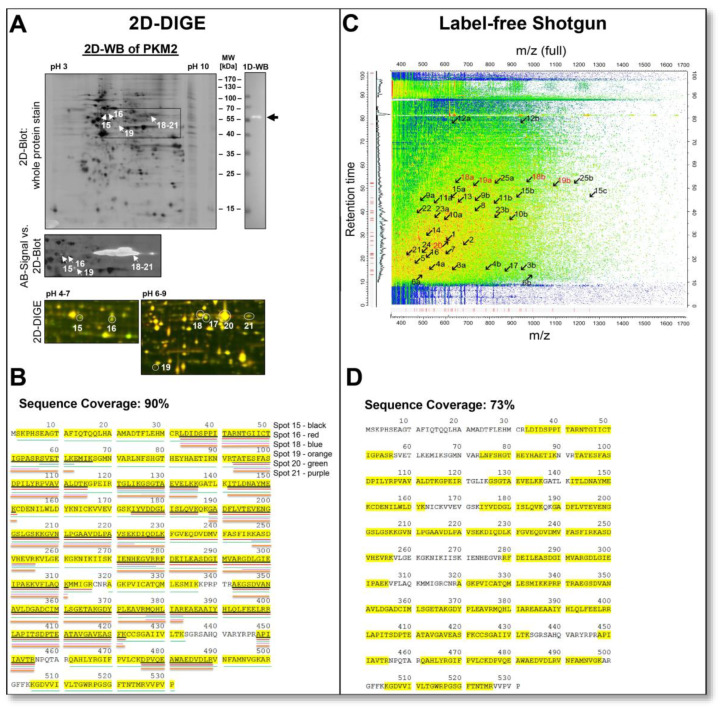
An unbiased proteoform exploration is only feasible with 2D-DIGE, as exemplified on PKM2. (**A**) PKM2-proteoform identification with 2D-DIGE: 30 µg Cy5-labeled DU145 protein was applied to IEF on a 7 cm pH 3–10 IPG-strip followed by 11.5% SDS-PAGE and blotted onto a PVDF membrane. The blotted Cy5 labeled proteins on the membrane were scanned with a laser scanner at 650 nm wavelength and shown as black spots. Afterwards, the blotted Cy5-labeled DU145 sample membrane is incubated with antibodies against PKM2 antibody (first left image). The specific signals are visualised by secondary HRP-conjugated antibodies and chemiluminescence and are shown as white spots. One-dimensional WB shows the PKM2 signal (white band) as the sum in a single band (right image). Overlay of Cy5-labelled protein spots (black spots) vs. PKM2 2D-WB signals (white spots), obtained through the Online Image Editor (https://www.online-image-editor.com, accessed on 25 July 2022). 2D-DIGE image shows the positions of different PKM2 proteoforms in the pH range 4–7 and pH range 6–9. The PKM2 spots are circled, and the spot number match those listed in Table 1. (**B**) Sequence coverage of PKM2 in 2D-DIGE/MS analysis. Yellow boxes represent the position of identified peptides in the protein sequence. (**C**) Representative centroid-peak view of all identified tryptic peptides and specific peptides assigned to PKM2. Proteotypic peptides are indicated in red. (**D**) Sequence coverage achieved by shotgun. Yellow boxes represent the position of identified peptides in the protein sequence. Abbreviations: 2D-DIGE—two-dimensional difference gel electrophoresis; MW—molecular weight; IEF—isoelectric focusing; IPG—immobilised pH gradient; WB—western blot.

**Figure 8 cells-12-00747-f008:**
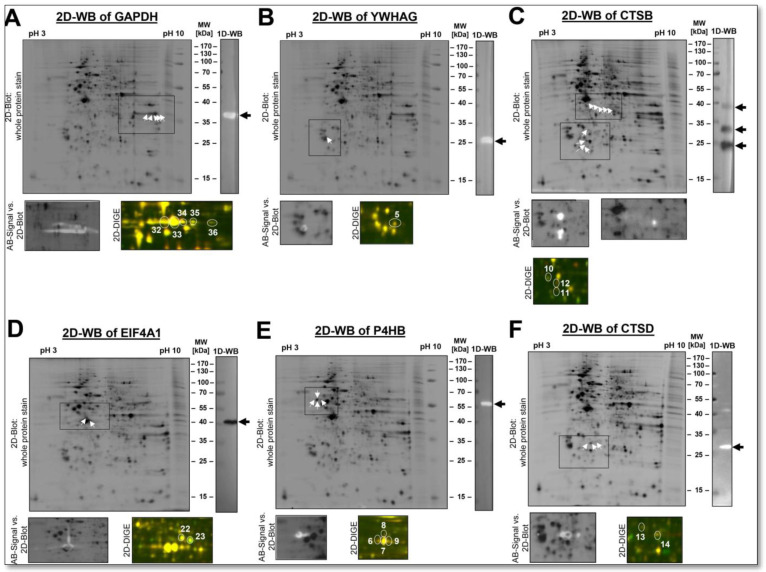
2D- and 1D-WB validation of selected proteins and proteoforms mutually identified by 2D-DIGE. 30 µg of Cy5-labeled DU145 protein was applied to IEF on a 7 cm pH 3–10 IPG-strip followed by 11.5% SDS-PAGE and blotted onto a PVDF membrane. These blotted Cy5 labeled proteins on the PVDF membranes were scanned with a laser scanner at 650 nm wavelength and shown as black spots. 2D-WB image (left large images) of DU145 sample. The PVDF membrane of the blotted Cy5-stained DU145 protein sample is stained with antibodies against (**A**) GAPDH, (**B**) YWHAG, (**C**) CTSB, (**D**) ElF4A1 (**E**) P4HB and (**F**) CTSD and the specific signals are visualised by secondary HRP-conjugated antibodies and chemiluminescence (white spots). Overlay of Cy5-labelled 2D-WB spot signals (black spots) vs. antibody-specific 2D-WB signals (white spots). One-dimensional WB shows the particular antibody signal (white band) as the sum in a single band (right image). The 2D-DIGE image shows the positions of different proteoforms (bottom), and the circled spots with a number match those listed in Table 1. The results of shotgun analysis from these proteins are shown in Appendix A. Abbreviations: 2D-DIGE—two-dimensional difference gel electrophoresis; MW—molecular weight; IEF—isoelectric focusing; IPG—immobilised pH gradient; WB—western blot.

**Figure 9 cells-12-00747-f009:**
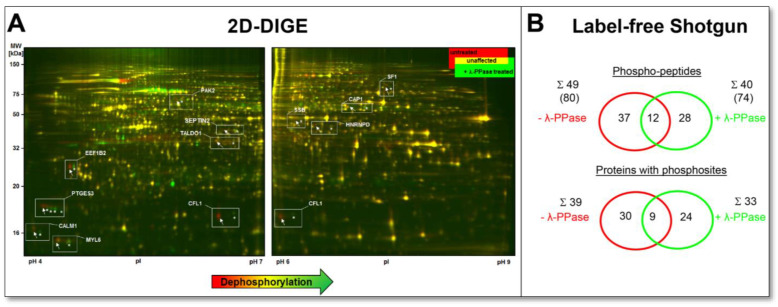
Characterization of phosphorylated proteoforms with 2D-DIGE and label-free shotgun. DU145 protein lysates were treated with ʎ-PPase to cleave the phosphate groups enzymatically from the amino acids S, T, and Y (STY) of the phosphorylated proteoforms. (**A**) Detection and identification of selected phosphorylated proteoforms by 2D-DIGE in DU145 lysates with ʎ-PPase treated (green) or not treated (red) spots. All these protein spots (proteoforms) were found with 100% reproducibility in all 2D gel runs. (**B**) Venn diagram summarising the overlap of identified phosphopeptides (top) and corresponding phosphoproteins (bottom) by “direct” shotgun proteomics without affinity-based enrichment. Top: detected, localised (*p* > 75%) and quantified phosphopeptides in untreated or ʎ-PPase treated DU145 protein lysates by the MaxQuant-Andromeda algorithm as summarised in the P(STY)-site table (numbers in parenthesis also include phosphopeptides without intensity). Source data and Supplementary Figures are summarised in Appendix A. Abbreviations: 2D-DIGE—two-dimensional difference gel electrophoresis; MW—molecular weight; pI—isoelectric point; ʎ-PPase—lambda phosphatase; *p*: localization probability.

**Figure 10 cells-12-00747-f010:**
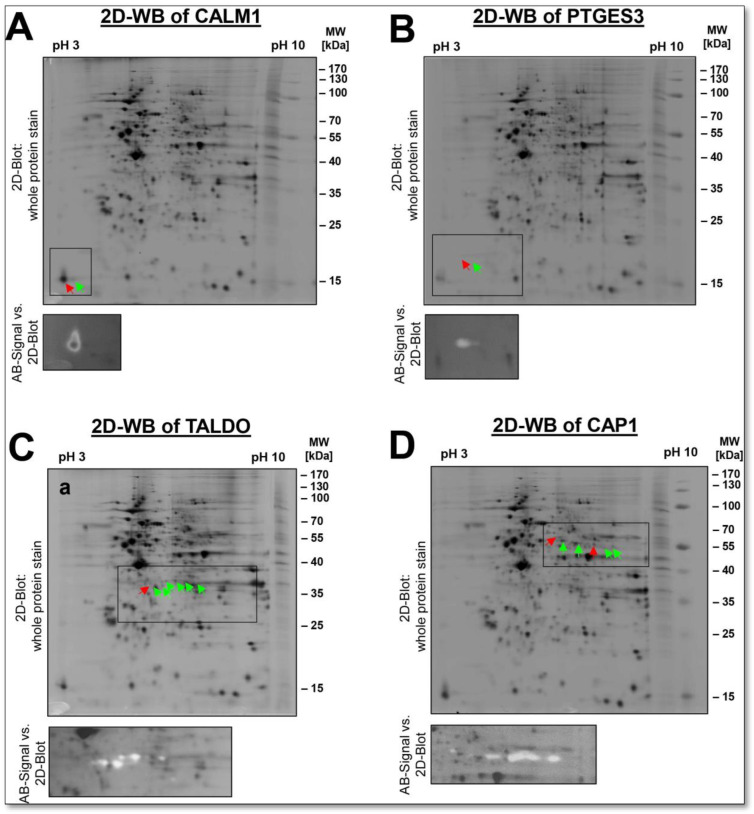
Two-dimensional WB confirmation of ʎ-PPase treatment sensitive proteins CALM, PTGES3, TALDO and CAP1. With IEF, 30 µg of Cy5-labeled DU145 protein lysate were separated on a 7 cm IPG strip, pH 3–10, followed by molecular weight separation by 11.5% SDS-PAGE and blotted onto a PVDF membrane. The 2D-WB membranes were scanned with a laser scanner at 650 nm wavelength and shown as black spots. The 2D-WB images of the Cy5-stained proteomes are stained with antibodies against (**A**) CALM1, (**B**) PTGES3, (**C**) TALDO and (**D**) CAP1 and the specific signals are visualised by secondary HRP-conjugated antibodies and chemiluminescence (white spots). Overlays indicate the particular Cy5- labelled spots in black spots and specific antibody signals with white spots. Red arrows indicate phosphorylated protein spots, and green arrows indicate non-phosphorylated protein spots. Abbreviations: 2D-DIGE—two-dimensional difference gel electrophoresis; IEF—isoelectric focusing; IPG—immobilised pH gradient; WB—western blot; MW—molecular weight; pI—isoelectric point; ʎ-PPase—lambda phosphatase.

**Table 1 cells-12-00747-t001:** Qualitative and Quantitative Proteoform to protein comparison of mutually identified entities by 2D-DIGE and shotgun.

Spot- No.	Gene Name	Protein Name	Identified by	pH Range	Practical pI	Theoretical pI	Practical MW [kDa]	Theoretical MW [kDa]	2D-DIGE				Label-Free Shotgun		2D-DIGE	Label-Free Shotgun
									CV_tech_ (%)	2D Spot Sum CVt_ech_ (%)	CV_total_ (%)	2D Spot Sum CV_total_ (%)	CV_tech_ (%)	CV_total_ (%)	Spot Size	LFQ- Intensity
1	CALM1	Calmodulin-1	2D-WB	4–7	3.66	4.09	14.1	16.8	3.9	33	12.9	34.2	15	55	9.7	218110164
2			MS	4–7	4.09		16.8		62.2		55.4				38.8	
3	PTGES3	Prostaglandin E synthase 3	MS	4–7	3.79	4.32	20.8	19.2	3.8	5.1	8.6	9.2	33.7	67.6	18.4	23571667
4			2D-WB	4–7	3.87		20.8		6.4		9.8				5.5	
5	YWHAG	14-3-3 protein gamma	MS	4–7	4.70	4.80	29.1	28.3	4.5	4.5	1.8	1.8	18.8	44.36	25.4	3275800
6	P4HB	Protein disulfide-isomerase	MS	4–7	4.72	4.76	58.3	57.1	3.3	4.5	3.7	5.5	8.9	63.2	12.7	63695001
7			MS	4–7	4.76		57.1		2.5		5.0				147	
8			MS	4–7	4.78		59.5		8.8		9.6				5.9	
9			MS	4–7	4.84		58.0		3.6		3.7				19.4	
10	CTSB	Cathepsin B	2D-WB	4–7	5.43	5.88	27.8	37.8	1.5	7.7	11.8	10.1	23.4	39.6	7.9	1219708
11			MS	4–7	5.53		26.1		14.4		5.1				3.4	
12			MS	4–7	5.54		27.0		7.1		13.5				5.6	
13	CTSD	Cathepsin D	MS	4–7	5.76	6.10	30.4	43.7	3.1	2.5	6.4	21.2	17.4	60.7	5.2	5884650
14			MS	4–7	5.97		28.8		1.9		36.1				24.1	
15	PKM2	Pyruvate kinase PKM2	MS	4–7	5.80	7.96	59.3	57.9	4.6	6.3	4.6	15.1	27.3	66.0	11.3	22507667
16			MS	4–7	6.08		58.3		3.7		5.3				17.8	
17			2D-WB	6–9	7.50		57.6		6.8		11.9				104.8	
18			MS	6–9	7.75		57.7		8.2		19.7				38.7	
19			MS	6–9	7.32		42.0		7.1		42.2				14.6	
20			MS	6–9	7.96		57.9		7.9		17.9				391.3	
21			MS	6–9	8.20		57.4		5.6		4.0				9.3	
22	EIF4A1	Eukaryotic initiation factor 4A-I	MS	4–7	5.81	5.32	48.6	46.2	3.0	2	13.2	11.8	11.0	48.8	80.6	41301834
23			MS	4–7	5.90		47.8		2.7		10.5				70	
24	TALDO1	Transaldolase	2D-WB	4–7	6.33	6.36	39.1	37.5	2.3	4.6	16.0	15.9	15.0	51.1	9.7	14659166
25			MS	4–7	6.82		38.0		4.2		14.4				19.1	
26			MS	4–7	7.32		39.2		7.4		16.9				20.3	
27	CAP1	Adenylyl cyclase-associated protein 1	MS	6–9	6.83	8.24	54.9	51.9	4.0	5.9	19.3	12.9	21.3	47.9	6.2	18092833
28			2D-WB	6–9	6.95		55.1		7.8		8.4				10.9	
29			2D-WB	6–9	7.14		55.5		5.5		17.1				8.0	
30			MS	6–9	7.38		54.8		4.4		11.2				21.4	
31			2D-WB	6–9	7.66		55.1		7.7		8.7				3.7	
32	GAPDH	Glyceraldehyde-3-phosphate dehydrogenase	MS	6–9	8.68	8.57	39.1	36.1	9.1	5.6	32.5	20.6	13.9	53.6	62.8	191046667
31			MS	6–9	8.78		39.0		7.7		25.1				626.3	
34			2D-WB	6–9	8.88		39.1		4.7		30.8				55.5	
35			2D-WB	6–9	9.01		39.0		3.1		5.3				13.0	
36			MS	6–9	9.21		38.8		3.3		9.5				5.6	

## Data Availability

Not applicable.

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
