# Peer review of "A Practical and Analytical Comparative Study of Gel-Based Top-Down and Gel-Free Bottom-Up Proteomics Including Unbiased Proteoform Detection"

_cells, 2023, doi:10.3390/cells12050747_

Round 1
Reviewer 1 Report (Previous Reviewer 1)
The authors have answered all the questions in a coherent and reasoned way, giving answers to everything asked.
The manuscript has been improved and its publication is recommended so that other groups can access this research.
Author Response
We sincerely thank Reviewer 1 for helpful revisions in optimising this proteomics study.
Reviewer 2 Report (Previous Reviewer 2)
thank you for the response,
again there are specific points that need to be answered:
1- the 1D Western blot should be provided in triplicate and should be provided as raw data in the supplementary data section, this has to do with common data element requirement
2-the proteins separated on the 2D- are already digested, thus what is referred as bopttom up. This isa fundemntal deviation from the proteomics literature. Uness you conform that the 2 D is a native gel , themn I would accept the naming of your method as top down !!!!!!!!!!
Author Response
Response to Reviewer 2 Comments (round two) (Attached file including some figures)
Reviewer 2: thank you for the response, again there are specific points that need to be answered:
Thank you for all your efforts in making this study more understandable for readers of Cells. We have implemented and answered your specific points, as you will read and see on the following lines and pages. In the manuscript, we have highlighted the textual changes of these revisions in yellow.
1- the 1D Western blot should be provided in triplicate and should be provided as raw data in the supplementary data section, this has to do with common data element requirement
Please excuse that we have not yet included these 1D Western blot triplicates in the Supplementary. We have now added all of our 1D Western blots in triplicate of the respective antibodies in the supplementary (Supplementary Figure S3 and at the end of this response letter). The descriptions of these 1D Western blots are added in method section (page 7 and line 338-350) and in the result section page 23 line 825-829) of the manuscript.
2-the proteins separated on the 2D- are already digested, thus what is referred as bopttom up. This isa fundemntal deviation from the proteomics literature. Uness you conform that the 2 D is a native gel , themn I would accept the naming of your method as top down !!!!!!!!!!
The two-dimensionally separated proteins and their proteoforms on a 2D gel are not digested! Undigested and intact proteins are separated and quantified using 2D electrophoresis. Before 2D separation, proteins are just unfolded and reduced, both to minimize protein-protein interaction and avoid complex formation.
Accordingly, this study's applied 2D electrophoresis method can detect and quantify intact proteins. However, no protein complexes can be analysed by this denaturation high-resolution 2D gel electrophoresis as can be done by native-native 2D gel electrophoresis [1].
Thus, in any case, we can detect differently abundant intact proteins and their intact proteoforms in a proteome with 2D gel electrophoresis.
Up to this point, the two-dimensional electrophoresis technology should conform to the classification of top-down proteomics [2-8].
Protein digestion to peptides is done at the very end of the pipeline. After 2-DE separation, spot quantification and selecting the spots of interest, those spots are cut out and undergo tryptic digestion for protein identification by MS.
However, these tryptic digestion steps are not necessary for all analytical 2D runs of a proteomics study; once a particular 2D spot has been identified in the respective proteome of a biological sample type and the preparation and separation methods are standardized, the respective 2D separated intact protein will be detectable in the same pI and MW region of the 2D gel.
One of our laboratory's most common applications is the human platelet proteome analysis from clinical samples, such as Alzheimer's or thrombosis patients. We already have 2D-DIGE gels from over five hundred patients in our 2D database. For these 2D analyses, it is only necessary to identify the respective protein spot one to three times with the MS; for all other following sample runs, the respective protein spot has its protein identification. Thus, after the reliable identification of a protein spot, the qualitative and quantitative analysis in the other samples should be top-down proteomics.
References
https://en.wikipedia.org/wiki/Top-down_proteomics#cite_note-:1-4
- Freeman, L. A. "Native-Native 2d Gel Electrophoresis for Hdl Subpopulation Analysis." Methods Mol Biol 1027 (2013): 353-67.
- Zhan, X., B. Li, X. Zhan, H. Schluter, P. R. Jungblut, and J. R. Coorssen. "Innovating the Concept and Practice of Two-Dimensional Gel Electrophoresis in the Analysis of Proteomes at the Proteoform Level." Proteomes 7, no. 4 (2019).
- Carbonara, K., M. Andonovski, and J. R. Coorssen. "Proteomes Are of Proteoforms: Embracing the Complexity." Proteomes 9, no. 3 (2021).
- Ohlendieck, K. "Top-Down Proteomics and Comparative 2d-Dige Analysis." Methods Mol Biol 2596 (2023): 19-38.
- Gelfi, C., and D. Capitanio. "Dige Analysis of Clinical Specimens." Methods Mol Biol 2596 (2023): 177-99.
- Marcus, K., C. Lelong, and T. Rabilloud. "What Room for Two-Dimensional Gel-Based Proteomics in a Shotgun Proteomics World?" Proteomes 8, no. 3 (2020).
- Naryzhny, S. "Inventory of Proteoforms as a Current Challenge of Proteomics: Some Technical Aspects." J Proteomics 191 (2019): 22-28.
- Kelleher, N. L. "Top-Down Proteomics." Anal Chem 76, no. 11 (2004): 197A-203A.
And finally, a more detailed technical description of 1D and 2D electrophoresis:
Both 1D and 2D gel electrophoresis represent classical top-down protein-separation methodologies, as intact proteins are separated based on their molecular weight (SDS-PAGE) or additionally based on their isoelectric point (first dimension of 2D). The peculiarity of 2D-gel electrophoresis is that before the first-dimensional run, intact proteins are first solubilized and denatured with high amounts of urea (usually 7M urea, 2M thiourea and the twitter ionic detergent 4% CHAPS). Before this IEF, 70 – 140 mM DTT is added to reduce disulfide bonds and break secondary and tertiary protein structures. After the separation by isoelectric point, the proteins on the immobilized pH gradient strip will be negatively charged by a brief wash with SDS. Thus separation occurs only according to their molecular weight in the electrical field, and proteins migrate from the cathode (-) to the anode (+). These protein preparation steps enable the separation of intact, denatured proteins based on their isoelectric point.
Graphical representation of top-down and bottom-up proteomics:
In case all these descriptions and references do not convince Reviewer 2 that 2D electrophoresis is a top-down proteomics method, we would change the title to “A practical and analytical comparative study of two common proteomics technologies including unbiased proteoform detection” and also remove the terms top-down and bottom-up proteomics from the manuscript, as these revisions would not significantly change the fundamental message of our study.
Figure S3. Validation of PKM2, GAPDH, YWHAG, CTSB, ElF4A1, P4HB, CTSD, CALM1, PTGES3, TALDO and CAP1 antibodies by one-dimensional Western blot analysis. (A) Ruthenium-based whole-protein stain of blotted proteins. Twelve µg of DU145 protein lysate was loaded per lane on an 11.5% SDS-PAGE with 28 sample wells and separated according to their molecular weight. The three biological DU145 replicates (BR-1, BR-2 and BR-3) of the previous proteomics study were alternately applied to both 1D gels with a pre-stained protein molecular weight marker in the MW range of 170 kDa - 15 kDa, and afterwards blotted onto two PVDF membranes. The apparently empty lanes contain this marker, which is visible to the naked eye but not with the device setting for the ruthenium stain. With visual guidance of the lanes with the separated protein molecular weight markers, the membranes were cut into 11 pieces, each containing the three biological DU145 replicates and one protein molecular weight marker. (B) These PVDF membranes were stained with antibodies against PKM2, GAPDH, YWHAG, CTSB, EIF4A1, P4HB, CTSB, CALM1, PTGES3, TALDO and CAP1, respectively. The specific signals were visualized by staining with the respective secondary HRP-conjugated antibody and with a substrate for chemiluminescence (left image of the respective protein). The signals were detected with the respective wavelength and filter settings of a UVP ChemStudio Imager. To check the correctness of the respective molecular weights of the different proteins detected, an overlay was made with the protein molecular weight marker (right picture of the respective Western blot).

Round 2
Reviewer 2 Report (Previous Reviewer 2)
Accept
This manuscript is a resubmission of an earlier submission. The following is a list of the peer review reports and author responses from that submission.
Round 1
Reviewer 1 Report
The paper named “A practical and analytical comparison of gel-based top-down 2 and gel-free bottom-up proteomics” is a very interesting paper were author indicate the main uses and the advantages and disadvantages from both 2D-page and Shotgun proteomic techniques. However this paper is very hard to read with a lot of information and comparisons between techniques that make difficult to understand the paper final objective.
Several questions may be answered by author
1) Material and methods author make the protein extraction in dry plates conserved at -80ºC. When author added the DIGE-buffer author make this in the freeze plate or allow plate reach the ambient temperature before add the buffer.
2) In point 2.3 author said that the first dimension voltage reach the 30kVh. Is this the voltage for the two different IPG-Strip (6-9 and 4-7). Please clarify. In the same sense the rehidratation step is the same in the two pH? This information is missing.
3) In point 2.5, trypsin among for the digestion is missing. The name or type of mass spectrometer is missing two.
4) In point 2.6 the author do not indicate the duration of the chromatography run and this date is very important because is not the same a gradient of 40 min that one of 90 or longer.
5) The label free technique that uses the author is based on? Is a SWATH? Spectral count please add this information in the material and methods
6) In result section in paragraph between line 411-418 author give information about spots and correlated this information with proteins however in one spots there can be more than one protein, and several spots can be the same protein. In this sense it is difficult make the comparison between spots in 2D vs proteins identified by Shotgun.
7) If author want compare 2D-PAGE and Shot gun why the proteins digestion was made using different enzyme?
8) In figure 2B author say that the in solution digestion is performed in 16 hours for trypsin and 2h for LysC, however in the methods section they say over night for both enzymes. Please clarify
9) In line 460 author say “this reason, this method has 464 largely replaced 2D technology today.” But the time cost of performing this technology is not the only reason to use other proteomic technology. Shotgun is able to identify not only the differential proteins but also the common proteins, and in most cases a lot of differential proteins more than 2D gels. 2D gels have the staining problem that can mask differential proteins.
10) In this paper nothing is said about the 2D DIGE staining that it is necessary to standardize and depend on the 2d DIGE buffers has the optimal conditions.
11) In paragraph between line 483 and 499 authors compare 2D with shotgun, but the interesting is not comparing. The interesting is using the two techniques to improve the results because the two techniques have a disadvantages and the two has advantages that can make a good idea use the two for obtain good results.
12) The shot gun replicates were made in different time. And the 2D_DIGE gels??
13) Regarding to the missing values in Shotgun techniques it is necessary to indicate that in 2D-gels there can be missing dates two that are those minoritary proteins that are under the staining limit of detection.
14) In line 553 author mentioned the label free technique pleases clarify this method. On the other hand author say that this technique has not internal standard however in some cases as SWATH analysis the retention time can be adjusted using the own protein peptides and this can be similar to have an internal standard.
15) In paragraph between line 731 and 737, make author a membrane digestion to identify the spots obtained in the 2D- western?
16) In the same paragraph author say that in figure 7A we can see 2D western however I think that this is a 2D gel not a western.
17) In figure 8 author shown 2D-wester how it is possible that a specific antibody give a lot of spots signal?
18) In lines 877 why author do not perform a Labelfree quantitative analysis to obtain the quantitative date of the proteoforms
19) In figure 9 author perform a PTM enrichment?
Reviewer 2 Report
we read with interest the technical article by Ercan et al comparing 2D-DIGE vs label-free LCMSMS analysis efficiency in protein coverage, PTM identification as well as the robustness of both techniques.
this work is of moderate interest for several reasons first of all, the use of 2D-DIGE is getting less common due to the cumbersome work and the need for experienced technical researchers as well as the time and cost they require.
Secondly, there are the TMT and iTRAQ labeling quantitative techniques that are more well-suited for biological samples compared to 2D-DIGE.
this kind of work has been already performed in literature extensively comparing 2DGEL vs label free ! No one in the field of proteomics considers one method as superior and being exhaustive for protein identification, each has its onewn limitation (a limitation section should be added)
I see the moderate novelty in this; however, it can contribute to the proteomics literature.
Comments:
the Western blot is not convincing as I see multiple spots in each gel and there is no way to indicate the specificity of the altered proteins
the study should indicate the contradictory proteins ie ones that show opposite changes between 2d-DIGE and label-free. are there three replicates present for each western blot, please provide the raw data. Additionally, a regular WB analysis using 1 D gel should be performed.
the work is using the term Top Down for the 2D-DIGE which is not correct as it is involving protein digestion and it is a bottom-up technique. (please modify)
the work needs English editing as sentences need correction